


# A rainfall-tracking travel time distribution model to quantify mixing and storage release preference in a large shallow lake by two-year stable isotopic data

Rong Mao[1], Xin Luo[1], Jiu Jimmy Jiao[1], Xiaoyan Shi[1], and Wei Xiao[2]

[1]Department of Earth Sciences, The University of Hong Kong, P. R. China.
[2]Yale-NUIST Center on Atmospheric Environment, International Joint Laboratory on Climate and Environment Change (ILCEC), Nanjing University of Information Science and Technology, Nanjing, Jiangsu Province, China

**Correspondence:** Jimmy Jiao (jjiao@hku.hk)

**Abstract.** Lake Taihu is the largest eutrophic lake in China that is shallow and connected to a dense river network. Severe eutrophication is frequently observed in Lake Taihu due to excess pollutant loadings. Understanding water cycle dynamics is essential for investigating this problem. The travel time distribution (TTD), residence time distribution (RTD) and storage selection (SAS) function describing how water is stored, mixed and released in the lake provide fundamental information on water cycle dynamics. In this study, a rainfall mixing model is established and coupled with the age master equation model to estimate the time-variant TTDs and RTDs of rainwater, river water and all water in Lake Taihu based on the two-year high-resolution isotopic data. In the rainfall mixing model, a novel rainfall mixing factor is introduced to quantify the mixing of rainwater with older lake water. The estimated range of travel time varies between 2-4 months and 7-9 months, depending on lake volume. Lake Taihu shows an inverse storage effect, i.e., the release preference shifts toward younger water when lake volume is large. The results of rainfall mixing model reveal distinct patterns and control factors between the TTDs of rainwater and river water, with rainwater contributing to 15%-25% of outflow. Then, the SAS functions of rainwater and river water are analyzed, which are controlled by different source zones and flow patterns. Temporal variation of spatial distribution of deuterium isotope composition illustrates storage release preference is controlled by the variation of horizontal flow paths and velocities in the lake.

## 1 Introduction

Water cycle dynamics are intimately related to biogeochemical and ecological processes in lakes. In this context, water cycle dynamics refers to temporal variations in lake volume, residence time, portion of water flowing through the lake, portion of water leaving the lake via evaporation, etc. Lake Taihu as the largest eutrophic shallow lake in China, suffers from severe and frequent cyanobacteria blooms. As Lake Taihu is an important drinking water source for the Yangtze River Delta urban agglomeration, such blooms may impact drinking water supply. In 2007, the drinking water supply was cut off for about 2 million people due to a harmful bloom in Lake Taihu (Qin et al., 2010). A series of efforts have been undertaken to improve the water quality in the lake, such as waste water treatment, sediment dredging, water division, etc. However, the lake water




quality has not been improved significantly after decadal investment and restoration (Qin, 2009; Qin et al., 2019). To understand the control factors of harmful algal blooms, studying the water cycle dynamics in Lake Taihu is an essential prerequisite for
investigating the biogeochemical and ecological processes in Lake Taihu.

The time-variant travel time distribution (TTD) and residence time distribution (RTD) which describe how the water is stored, mixed and released in a system provide fundamental information about the timescale of water transport, storage dynamics and variation of flow regimes. The travel time of a water parcel is defined as the time-interval between its entering and leaving time in a system; and the residence time of a water parcel is defined as the duration for which it remains within a system subsequent
to a certain input event, such as precipitation, river inflow to the lake, etc. The TTD and RTD are then introduced to quantify the age distribution of water leaving the system and the age distribution of water residing in the system, respectively (Botter et al., 2011; McGuire and McDonnell, 2006). The theory of time-variant TTD and RTD has been widely used to explore the catchment scale water and solute transport in aquifers (Benettin et al., 2017; Danesh-Yazdi et al., 2017; Yang et al., 2018). In this theory, water parcels within a hydrological control volume are labeled with their respective ages, and the StorAge Selection
(SAS) function is utilized to establish the connection between the TTD and RTD of a system. The SAS function characterizes the storage 'selection' preference, specifically the preference for releasing water parcels of different ages from storage to outflow (Botter et al., 2011; Harman, 2019; van der Velde et al., 2012). In Lake Taihu, both the travel time and residence time can help understand water and biogeochemical cycles in the lake so, the time-variant TTD, RTD and SAS function of Lake Taihu are evaluated.

The applications of TTD theory in many previous studies (Benettin et al., 2015, 2017; Danesh-Yazdi et al., 2017) indicate the transport of water parcels and the time variability of the system can be well described by TTD, RTD and SAS function. Kim et al. (2016) further decomposed the time variability into two parts: (1) the internal variability associated with changes in partitioning between flow paths, (2) the external variability driven by the variations of external forces such as precipitation. The internal variability can be captured by the SAS function; and the external variability is reflected in the time-variant TTD.
However, situations may be different for the time variability in the lake. In previous studies of TTD in catchments, rainfall is the only water source, so the TTD, RTD and SAS function are determined by the transport of rain water; but in Lake Taihu, the inflow consists of river and precipitation, leading to the time variability influenced by the transport and mixing of both rain and river water. Specifically, the different entry points for rain and river water results in distinct flow patterns between the rain and river water, consequently impacting the spatial distribution of water age in the lake. This finally determines the
time variabilities of TTD, RTD, and SAS functions of the lake water. To the best knowledge of the authors, there is only one study (Smith et al., 2018) which applied the TTD theory to lakes; and in previous TTD models (Benettin et al., 2015, 2017; Danesh-Yazdi et al., 2017), the impact of diverse flow patterns from different sources on the time variabilities of TTD, RTD, and SAS functions has not been investigated yet. Smith et al. (2018) assessed the TTDs in two lakes: one lake has an inflow river and an outflow river with rainfall contributing to 1/4 of the total input, the other has two inflow rivers and an outflow river
with rainfall contributing to 1/42 of the total input. In the study of Smith et al. (2018), a single SAS function for mixed lake water is employed. The determination of how the mixing of rain water and river water, with their distinct entry points and flow patterns, affects the outflow preference of lake water remains unknown. Therefore, a rainfall-tracking TTD model is highly in





need to resolve the mixing of multiple sources. In Lake Taihu, where rainfall accounts for approximately 1/3-1/4 of the total input, the entry points of rain water span across the lake surface, presenting different entry points compared to those of river

water. With the result of the rainfall-tracking TTD model, the following key scientific questions related to TTD in large shallow lakes can be addressed:

(1) what are the characteristics of TTDs, RTDs and SAS functions of river water and rain water?

(2) What are the relations among TTDs of river water, rain water and all water?

(3) How does the transport of river water and rain water determine the time variability of storage release preference of the

lake?

Furthermore, distinguishing the TTDs and RTDs of rain water and river water is essential for investigating the biogeochemical processes within the lake due to the chemical differences between these water sources. Specifically, several previous studies in Lake Taihu (Wang et al., 2019; Xu et al., 2021) have investigated the impacts of nutrient dynamics on the phytoplankton growth using the nutrient-driven dynamic eutrophication model. Li et al. (2011) discussed the relation between water age and

the algal concentrations in the lake. In these studies, the nutrient contributions of rainfall and inflow rivers are different significantly. Therefore, the estimation of TTDs, RTDs and SAS functions of rain water and river water will provide fundamental information for the study of the impact of mixing of multiple sources and water age distribution on the phytoplankton growth in Lake Taihu.

Different from the computational fluid dynamics modeling of water age in Lake Taihu (Li et al., 2011), the TTD and RTD

obtained in our study are inferred from stable isotope transport. Li et al. (2011) simulated the spatial distribution of water age in Lake Taihu and investigated the influence of wind and water transfer project on the spatial distribution of water age. However, the three scientific questions related to the water age introduced earlier in Lake Taihu remains unclear. The results of the application of the proposed TTD model in our study will provide some new insights into the water cycle dynamics in Lake Taihu as well as their ecological impact.

In this study, time-variant TTDs, RTDs and SAS functions of rain water, river water and all water are evaluated based on the two-year time-series stable isotope data. Firstly, the mass balance model of the lake water is constructed; then, the age master equation model tracks the volume of water parcels of different ages; and the isotope mass balance model is employed to calibrate the model parameters used in the age master equation model. Besides, the rainfall mixing model is introduced in this study to differentiate the transport of rain water and river water in the lake, so that the TTDs, RTDs and SAS functions of rain

water and river water can be evaluated. Finally, the spatial distributions of stable isotope content in Lake Taihu are analyzed to illustrate the control factors of the storage release preference of water in Lake Taihu.

## 2  Study area and data

Lake Taihu is located in south of Yangtze River Delta in eastern China. It is a large shallow lake with a surface area of 2400 km$^2$ approximately and with a mean depth of 1.9 m. The catchment of Lake Taihu is highly urbanized and has a dense population.

The area of this catchment is about 36500 km$^2$. The north boundary of the catchment is Yangtze River, and part of the south



boundary is Qiantang River. The lake is connected to a dense river network and is dammed by many dikes surrounding the lake to regulate water inflow and outflow. Generally, the inflow rivers are mainly located at the western mountainous area and the north area, and the outflow rivers are located at the eastern area. The catchment is under subtropic climate condition which is characterized by high temperature and precipitation in summer and low in winter.

The catchment of Lake Taihu is divided into 7 sub-basins which are shown in Fig. 1, and five of the sub-basins are connected to Lake Taihu directly. Sub-basins ① and ② shown in Fig. 1 in the west of the catchment are the two sub-basins which contribute the most surface water to Lake Taihu.

The inflow runoff and outflow runoff in all sub-basins connected to Lake Taihu are monitored continuously by the Bureau of Hydrology, Taihu Basin Authority of Ministry of Water Resources, China. The flow rate data of the surface runoff, lake
volume data and precipitation data are obtained from the *Monthly Hydrological Report* (The Bureau of Hydrology, Taihu Basin Authority of Ministry of Water Resources, China., 2014) for the period from November 2012 to December 2014 (see Supplementary Information (SI)). Generally, sub-basin ① is dominated by river inflow, sub-basins ③ and ④ are dominated by river outflow, and the inflow and outflow rates at sub-basin ② are of the same magnitude. The river inflow to Lake Taihu and outflow from Lake Taihu at sub-basin ⑤ are mainly regulated by the water diversion project from Yangtze River. The total
inflow to Lake Taihu is 8.963 $\times 10^9$ m$^3$ in 2013 and 10.217 $\times 10^9$ m$^3$ in 2014; and the total outflow from Lake Taihu is 8.491 $\times 10^9$ m$^3$ in 2013 and 10.249 $\times 10^9$ m$^3$ in 2014.

The overall precipitation is 1067.9 mm in 2013 and 1232.70 mm in 2014. The wet season of Lake Taihu is usually from May to September. The large precipitation in October 2013 is due to the influence of typhoon from October 6th to 8th.

The temperature of Lake water, relative air humidity and the wind speed at 10 m above the lake are monitored daily along
with the continuous monitored stable isotopes ($^2$H and $^{18}$O) of lake water (Xiao et al., 2016). Obviously, the temperature of lake water and relative humidity are higher in summer season. In this study, evaporation of lake water is evaluated based on these three quantities which are presented in SI.

Fig. 2 shows the locations of sampling points for stable isotopes including deuterium ($^2$H) and oxygen-18 ($^{18}$O). There are 51 rivers surrounding Lake Taihu which are sampled seasonally during the year of 2013 and 2014. 29 lake water samples
were collected seasonally from 2011 to 2014, and lake water samples were collected daily in one station from 2012 to 2014. The precipitation for deuterium analysis was sampled continuously from October 2012 to July 2015 at the north boundary of Wuchengxiyu sub-basin (Xiao et al., 2016). As the elevation difference between the sampling point for precipitation and the surface of Lake Taihu is negligible, the isotopic lapse rate in precipitation is insignificant. Therefore, the isotope contents in precipitation in the monitoring station can represent the isotopic compositions in the precipitation above the entire lake area.
These intensive monitored isotope data will be used to calculate the time-variant TTD in this study. Detailed stable isotope data in Lake Taihu and rivers are presented in SI.





**Figure 1.** The catchment of Lake Taihu with 7 sub-basins: ① Huxi sub-basin; ② Zhexi sub-basin; ③ Hangjiahu sub-basin; ④ Yangchengdingmao sub-basin; ⑤ Wuchengxiyu sub-basin; ⑥ Puxi sub-basin; ⑦ Pudong sub-basin. Yangtze River is located at the north boundaries of sub-basins ①, ④, ⑤, ⑥ and ⑦.

## 3   Methodology

The method to calculate time-variant TTDs, RTDs and SAS function of rain water and river water is developed by coupling the age master equation model (Botter et al., 2011) and rainfall mixing model. Generally, the evaluation of time-variant TTD
by the age master equation model includes three steps: (1) establish the mass balance model of lake water; (2) establish the age





**Figure 2.** Sampling points of lake water and river water. Red star: the daily sampling point of lake water; red dots: the seasonal sampling points of lake water; lines connected to the lake: the sampling rivers.



master equation model and isotope transport model; (3) evaluate the age-ranked storage by calibrating the age master equation model and isotope transport model with the isotope data in lake water. However, the transport of rain water and river water in the lake cannot be tracked by the three steps, so the rainfall mixing model is developed in this study and is incorporated with the age master equation model at the second step.

## 3.1 Mass balance model for lake water

The mass balance model of lake water in this study describes the change of lake volume (assuming constant water density) in response to precipitation, river inflow, river outflow, and evaporation of lake water. It can be expressed as:

$$\frac{\mathrm{d}S(t)}{\mathrm{d}t} = F_{in}(t) - F_{out}(t) + (P(t) - E_L(t)) \cdot Area \tag{1}$$

where $S(t)$ is the volume of Lake Taihu at time $t$; $F_{in}(t)$ and $F_{out}(t)$ represent the total flow rate of inflow rivers and total
flow rate of outflow rivers at time $t$ respectively; $P(t)$ is the precipitation rate at time $t$; $E_L(t)$ is the evaporation rate at time $t$; $Area$ is the surface area of Lake Taihu. The fluctuation of surface area of Lake Taihu in response to water level is ignored due to the significantly large size of the lake.

The precipitation $P(t)$ and lake volume $S(t)$ are presented in SI. The total flow rate of inflow rivers $F_{in}(t)$ and total flow rate of outflow rivers $F_{in}(t)$ are calculated using the excerpted flow rate data of the five sub-basins. The only unknown quantity
is the evaporation rate $E_L(t)$ of Lake Taihu.

The evaporation rate of open lake water is estimated by Penman equation (Penman, 1948):

$$E = 0.033 \times (0.78u_{10})^{0.68}(e_s - e_d) \tag{2}$$

where $u_{10}$ is the wind velocity (miles/day) at 10 m above the evaporating lake surface; $e_s$ is the vapour pressure (unit: mm Hg) at the evaporating surface of the lake; $e_d$ is the vapour pressure (unit: mm Hg) in the atmosphere above the lake surface. The
vapour pressure $e_s$ at the evaporating surface is determined by the temperature of water at this surface (Buck, 1996):

$$e_s = 0.61121 \times e^{\left(18.678 - \frac{T}{234.5}\right)\left(\frac{T}{257.14 + T}\right)} \tag{3}$$

where $T$ is the temperature of water (°C), and the unit of $e_s$ in Eq. (3) is kPa. $e_d$ in Eq. (2) is determined by the saturated vapour pressure and relative humidity:

$$e_d = e_s \cdot \mathrm{RH} \tag{4}$$

where RH is the relative humidity in the ambient air above the lake. Therefore, the evaporation rate at the monitoring point can be estimated by combining Eqs. (2)-(4) using the quantities introduced previously including temperature of lake water, relative humidity and wind speed at 10 m above the lake.

Due to potential differences in meteorological measurements (RH, $T$, $u_{10}$) between the lake shore and areas beyond the lake shore, and considering that the Penman equation (Eq. (2)) is derived empirically from a water tank (0.76 m in diameter),
using this equation directly to estimate evaporation across the entire expanse of a large lake may result in a biased estimation





(McCUEN and ASMUSSEN, 1973; Finch and Hall, 2001). This bias is evidenced by substituting the estimated evaporation

rate using Eq. (2) to the mass balance equation for lake water (Eq. (1)), which leads to a decreases in the lake volume over
time (see SI). As a result, similar to the method used in the prior study (Smith et al., 2018), a constant calibration parameter $\kappa$
is introduced:

$$E_L = \kappa \cdot E_0 \tag{5}$$

where $E_L$ is the calibrated mean evaporation rate of the entire lake surface, and $E_0$ is the estimated evaporation rate using Eqs.
(2)-(4). The calibration parameter $\kappa$ is estimated by minimizing the difference between the observed and the calculated lake
volumes based on the mass balance equation for lake water (Eq. (1)). The calibration parameter $\kappa$ is yielded to be 0.7879. This
parameter provides a scaled estimation of evaporation using the Penman equation, while preserving the temporal fluctuations
of the evaporation flux across the lake.

### 3.2 The rainfall-tracking travel time distribution model

The rainfall-tracking travel time distribution model consists of two sub-models: the age master equation model for all lake
water and the rainfall mixing model. The two sub-models will be introduced in this subsection separately. After this, how the
two sub-models are coupled to calculate the TTDs, RTDs and SAS functions will be illustrated.

#### 3.2.1 Age master equation model for all lake water

The age master equation model was first introduced by Botter et al. (2011). It differentiates water parcels in storage by ages
and tracks their volume over time. The volume of the aged water parcels is quantified as age-ranked storage $S_T(t,\tau)$ (Harman,
2015). The age-ranked storage is defined as the volume of water parcels in storage with age that is younger than or equal to
$\tau$ at time $t$. Let $S(t)$ represent the total volume of water in storage at time $t$, then the age-ranked storage is a function of total
storage and cumulative RTD:

$$S_T(t,\tau) = S(t)P_s(t,\tau) \tag{6}$$

where $P_s(t,\tau)$ is the cumulative RTD with age $\tau$ at time $t$. As the water parcels from different sources are not differentiated in
the age master equation model, the age ranked storage $S_T(t,\tau)$ is the total volume of rain water and river water aged younger
than or equal to $\tau$ at time $t$.

In Lake Taihu, the change of age-ranked storage is determined by precipitation, river inflow, river outflow and evaporation.
The precipitation and river inflow supply new water to Lake Taihu with the assumption that water entering the lake has an age
of zero. The river outflow and evaporation take the water out of the lake. The compositions of water parcels of different ages
in outflow rivers and evaporation are characterized by the backward TTDs for outflow rivers and evaporation respectively. The
cumulative form of the age master equation is employed to describe the evolution of age-ranked storage in time:

$$\frac{\partial S_T(t,\tau)}{\partial t} + \frac{\partial S_T(t,\tau)}{\partial \tau} = J(t) + F_{in}(t) - F_{out}(t)\Omega_Q(t, S_T(t,\tau)) - E(t)\Omega_E(t, S_T(t,\tau)) \tag{7}$$





where $J(t)$ and $E(t)$ are the rainfall input volume and evaporation volume across the lake, i.e., $J(t) = P(t) \cdot Area$ and $E(t) = E_L(t) \cdot Area$; $\Omega_Q(t, S_T(t,\tau))$ and $\Omega_E(t, S_T(t,\tau))$ are the cumulative age-ranked SAS functions for river outflow and evaporation respectively. The terms $F_{out}(t)\Omega_Q(t, S_T(t,\tau))$ and $E(t)\Omega_E(t, S_T(t,\tau))$ quantify the contribution of water in $S_T(t,\tau)$ to river outflow and evaporation respectively. As Lake Taihu is connected to a dense river network, the evaluation of age-ranked SAS function for each outflow river will introduce numerous shape parameters. This study considers only one SAS function for all outflow rivers, sufficiently revealing the transport dynamics of lake water, and utilizes one TTD to characterize the travel time of water in all outflow rivers. Fig. 3 (a) illustrates the evolution of age-ranked storage. This evolution is described by the age master equation (Eq. (7)). The evolution of age-ranked storage shown in Fig. 3 (a) is implemented using the Euler-forward scheme described by Benettin and Bertuzzo (2018). The detail of the implementation is presented in SI.

In the age master equation (Eq. (7)), the cumulative age-ranked SAS function $\Omega(t, S_T(t,\tau))$ is an essential function determining how the water is released from lake storage via river outflow and evaporation. The cumulative age-ranked SAS function for evaporation is assumed to be cumulative uniform distribution:

$$\Omega_E(t, S_T(t,\tau)) = \frac{S_T(t,\tau)}{S(t)} \tag{8}$$

The cumulative uniform distribution for evaporation means that the age composition of lake water is the same as that of water vapour above the lake surface. This SAS function is employed based on the unstratified nature of lake water in the vertical direction and the negligible spatial variation of evaporation across the lake surface. The uniform vertical distributions of temperature and deuterium isotope support the unstratified and vertical well-mixing nature of lake water(Xiao et al., 2016). Notably, prior studies in Lake Taihu (Wang et al., 2014, 2019) have shown a uniform all-way radiation pattern across the lake, with evaporation primarily influenced by radiation energy rather than other biophysical driver (wind speed, water depth, water pollution status). Their study concluded that the spatial variation of evaporation is minimal. Consequently, the uniform SAS function for evaporation assumes that all water within the lake has an equal probability leaving the lake via evaporation.

With reference to previous TTD studies (van der Velde et al., 2012; Yang et al., 2018), the beta distribution provides an easy and robust approach to characterize SAS function. Then, the cumulative beta distribution is employed to characterize the cumulative age-ranked SAS function for river outflow:

$$\Omega_Q(t, S_T(t,\tau)) = \frac{B(\frac{S_T(t,\tau)}{S(t)}; \alpha, \beta)}{B(\alpha, \beta)} \quad (\alpha > 0, \beta > 0) \tag{9}$$

where $B(\frac{S_T(t,\tau)}{S(t)}; \alpha, \beta)$ is the incomplete beta function with respect to $\frac{S_T(t,\tau)}{S(t)}$ which is the cumulative RTD of water in the lake; $B(\alpha, \beta)$ is the beta function; $\alpha$ and $\beta$ are the two positive shape parameters determining the shape of cumulative beta distribution and the storage 'selection' preference of Lake Taihu. For example, if $\beta > \alpha > 1$ or $\alpha < 1, \beta \leq 1$, the storage prefers releasing young water; if $\alpha > \beta > 1$ or $\alpha \leq 1, \beta < 1$, the storage prefers releasing old water; if $\alpha < 1, \beta < 1$, the storage prefer releasing both young and old water. The determination of shape parameters on storage 'selection' preference are presented in SI.

In an unsteady state system, the age-ranked storage selection function may vary with time, indicating the shape of beta distribution changes with time. This study assumes that $\alpha$ is a constant and $\beta$ is a function of lake volume $S(t)$ which varies in





time:

$\alpha = e^{a}; \quad \beta = e^{b+k \cdot w(t)}$  (10)

where $\alpha$ and $\beta$ are both expressed as an exponential function to ensure them to be always positive; $w(t)$ is the normalized lake water storage $w(t) = \frac{S(t)-S_{min}}{S_{max}-S_{min}}$. Consequently, the parameter $k$ in equation (10) can be employed to delineate the inverse and direct storage effects. If $k$ is positive, the shape parameter $\beta$ increases with lake volume $S(t)$, so the lake prefers releasing young water when lake volume is high. This is known as the inverse storage effect (Harman, 2015). If $k$ is negative, the shape

parameter $\beta$ decreases with lake volume $S(t)$, so the lake prefers releasing old water when lake volume is high. This is known as the direct storage effect.

In summary, the age master equation model for all lake water tracks the transport of water parcels of different ages from input to output. To further differentiate the TTDs and SAS functions of rain water and river water, the rainfall mixing model is introduced.

### 3.2.2   Rainfall mixing model in Lake Taihu

(Note: this section is quite new and may be tough, I am glad to answer any questions in the comments. I will delete this sentence, if published.)

Rainfall mixing model is established based on the transport of river water and rain water in Lake Taihu. As Lake Taihu is

a large shallow lake with mean depth of 1.9 m and the stratification of lake water is negligible, the flow direction of the lake is dominated by horizontal flow (Fig. 4 (a)). Therefore, the path lines of river water parcels spread horizontally from inflow to outflow rivers, and the rain water parcels follow the same path lines as the river water parcels. The main difference between them lies in their starting points, with the path lines of rain water starting at the entry points on the lake surface.

Based on the unstratified and vertical well-mixing nature of lake water introduced after Eq. (8), it can be inferred that, at

the entry points of rain water, the new rain water parcels (age 0) are well-mixing vertically with the lake water parcels. For simplicity, these lake water parcels are referred to as the local lake water parcels. The vertical mixing of rain and river water is illustrated in Figs. 4 (a) and (b). In Fig. 4 (a), the transport of lake water is decomposed into two parts: the horizontal flow and the vertical mixing process. The transport of river water in the lake is determined by horizontal flow, and is not well mixed with the other lake water due to the spatial variation of horizontal flow; while the mixing of rain and river water in the lake is

controlled by the vertical well mixing process due to the unstratified nature of lake water.

Before introducing rainfall mixing model, the definitions of age-ranked storage complement and age-ranked river outflow complement should be clarified. The age-ranked storage complement $\overline{S_T}(t,\tau)$ is the volume of water of age older than $\tau$ in storage at time $t$ (see Fig. 3 (b)); and the age-ranked river outflow complement $\overline{F_{out_T}}(t,\tau)$ is volume of water older than $\tau$ in outflow rivers at time $t$. The idea to introduce these complement quantities is based on the fact that the mixed lake water is

always older than the mixed rain water, since rain falls directly onto the lake surface.





Then, the rainfall mixing model is developed based on the vertical mixing process in Lake Taihu. Fig. 4 (b) provides a detailed depiction of the temporal evolution of age composition in a flow tube. In Fig. 4 (b), the rain water that enters the lake surface during the time period between t=0 and t=1 is well-mixed vertically with the inflow river water and the local lake water. Assuming the rainfall rate is uniform along the flow tube and the spatial variation of water depth is negligible in this shallow lake, the rainfall mixing factor (denoted as $f(t = 1, \tau = 0)$, $t$ is time, $\tau$ is age) is defined as the ratio of rainfall depth to the lake water depth, which is also equal to the the ratio of the volume of rain water entering during t=0-1 to the total mixed volume.

At the time $t = 1$, the mixed water is separated into three components: water in the lake, in outflow rivers and evaporated into the air, as depicted in Fig. 4 (b). Since the rain water entering during t=0-1 has been well-mixed with local lake water and subsequently flow with the lake water horizontally, these lake water and rain water can be considered as a unified entity with a fixed ratio of rain water all the time, i.e., $f(t_i, \tau_1) = f(t_i + (\tau_2 - \tau_1), \tau_2)$, whether in the lake or in outflow rivers. This fixed ratio is the rainfall mixing factor defined earlier. Therefore, the rainfall mixing factor remains constant within this unified entity all the time (for example: $f(t = 2, \tau = 1) = f(t = 1, \tau = 0)$) persisting both in the lake and in outflow rivers. The volume of the unified entity equals the age-ranked storage complement $\overline{S_T}(t, \tau)$ in the lake and the age-ranked river outflow complement $\overline{F_{out_T}}(t, \tau)$ in outflow rivers. Then, the volume of rain water from the period t=0-1 in the lake is determined by: $s_{rain}(t = 2, \tau = 1) = \overline{S_T}(t = 2, \tau = 1) \cdot f(t = 2, \tau = 1)$. Similarly, the volume of rain water from 0-1 in outflow rivers $F_{out\_rain}(t = 2, \tau = 1)$ can be calculated as $\overline{F_{out_T}}(t = 2, \tau = 1) \cdot f(t = 2, \tau = 1)$. In this way, the volume of rain water remaining in the lake can be tracked by coupling the rainfall mixing model and age master equation model.

It should be clarified that the rainfall mixing factor is fixed for water entering at the same time, due to the unstratified and vertical well-mixing nature of lake water; but the factor can still change in time; time and age are two independent variables. Specifically, if we track the changes of volumes of rain water and river water entering at a certain time $t_i$, the ratio of volumes of rain water entered at $t_i$ to lake water entered at $t_i$ is fixed when their ages get increased, i.e. $f(t_i, \tau_1) = f(t_i + (\tau_2 - \tau_1), \tau_2)$. However, if we compare the rainfall mixing factors between two different rainfall events, the rainfall mixing factors can be different.

In general, for a specific rainfall event occurring during $[t - \tau, t - \tau + \Delta t)$, the rainfall mixing factor is defined as:

$$f(t, \tau) = \frac{J(t - \tau) \cdot \Delta t}{S(t - \tau) + F_{in}(t - \tau) \cdot \Delta t + J(t - \tau) \cdot \Delta t} \tag{11}$$

The rainfall mixing factor is expressed in discretized form for the period $[t - \tau, t - \tau + \Delta \tau)$. During this period, the rainfall volume is $J(t - \tau) \cdot \Delta t$ (Unit: m$^3$), and the river inflow volume is $F_{in}(t - \tau) \cdot \Delta t$ (Unit: m$^3$). The storage volume at the time $t - \tau$ is $S(t - \tau)$ (Unit: m$^3$). It should be noted that the river outflow volume is not included in Eq. (11), this is because the outflow occurs after the mixing process in the lake. Moreover, the volume of rain water in the lake and outflow rivers from this rainfall event can be tracked as:

$$s_{rain}(t, \tau) = \overline{S_T}(t, \tau) \cdot f(t, \tau) \tag{12}$$

$$F_{out\_rain}(t, \tau) = \overline{F_{out_T}}(t, \tau) \cdot f(t, \tau) \tag{13}$$

As shown in Eqs. (12) and (13), rainfall is well-mixed only with lake water older than itself. This differs from the well-mixing assumption in previous TTD studies (Benettin et al., 2013; Hrachowitz et al., 2013, 2015), where water of a specific





285 age was assumed to be well-mixed with all water in storage. In fact, the TTD model with vertical well mixing is proposed for the first time in this study. It is not a truly well-mixing model, but selectively mixing of rain water and the lake water older than rain water. This is also why the age-ranked storage complement and age-ranked outflow complement are introduced in the rainfall mixing model.

In summary, in the rainfall mixing model, the rainfall mixing factor quantifies the mixing ratio of volume of the rain water 290 to the lake water with age older than the rain water. This factor is constant for rain water from the same rainfall event; but for rain water from different rainfall event, the factor may be different depending on the ratio of rainfall depth to lake water depth. Therefore, this factor can be used to track the volumes of rain water from the same event. Moreover, all the other mixing processes in the model are not well-mixed processes, such as the mixing of young rain water and old rain water in the lake, the mixing of young river water and old river water in the lake, and they are quantified by the SAS functions of rain water and 295 river water respectively.

### 3.2.3 Incorporation of rainfall mixing model with the age master equation model

At this stage, the rainfall mixing model is incorporated with the age master equation model to separate TTD of all water into TTDs of rain water and river water. As the age-master model and rainfall mixing model are two independent models, two constraints and one coupling equation are introduced as follows to ensure the results of the two models are not conflict and to 300 couple the age-master equation model and rainfall mixing model.

Constraint 1: $\qquad s_{rain}(t,\tau) = \overline{S_T}(t,\tau) \cdot f(t,\tau) \leq s(t,\tau)$ (14)

Constraint 2: $\quad F_{\text{out}\_rain}(t,\tau) = \overline{F_{\text{out}\,T}}(t,\tau) \cdot f(t,\tau) \leq F_{\text{out}}(t) \cdot \overleftarrow{p_Q}(t,\tau)$ (15)

Coupling equation: $\qquad \overline{F_{\text{out}\,T}}(t,\tau) = F_{\text{out}}(t)\left(1 - \Omega_Q(t, S_T(t,\tau))\right)$ (16)

Eq. (14) ensures that the calculated volume of rain water aged $\tau$ in the lake, i.e., $s(t,\tau)$, is always smaller than the volume of 305 total lake water aged $\tau$. Eq. (15) ensures that the calculated volume of rain water aged $\tau$ in river outflow is always smaller than the volume of total water aged $\tau$ in river outflow. Eq. (16) links the constraint 2 with the cumulative age-ranked SAS function. The two constraints are based on the fact that both the lake water and water in outflow rivers consist of water from rain fall and from river inflow.

The conflict of the two models usually occours in the case: if improper cumulative age-ranked SAS function $\Omega_Q(t, S_T(t,\tau))$ 310 is applied to the age master equation model, the calculated $\overline{S_T}(t,\tau)$ and $\overline{F_{\text{out}\,T}}(t,\tau)$ from the age master equation model will violate the two constraints. This is usually caused by the underestimation of the storage release preference of rain water using the traditional age-ranked SAS function (Harman, 2015). Therefore, an improved cumulative age-ranked SAS function $\Omega_Q(t, S_T(t,\tau))$ is developed to characterize the storage release preference considering the rainfall mixing process. The procedures to calculate the improved cumulative age-ranked SAS function $\Omega_Q(t, S_T(t,\tau))$ are as follows:

(1) Given certain shape parameters $(\alpha, \beta)$, calculate the cumulative beta distribution $\Omega_{Q_0}(t, S_T(t,\tau))$ and backward TTD $\overleftarrow{p_{Q0}}(t,\tau)$ using Eq. (9);





(2) At each time step, check whether $\Omega_{Q_0}(t, S_T(t,\tau))$ and $\overleftarrow{p_{Q0}}(t,\tau)$ satisfy the constraint 2 for all age $\tau$. If not, the storage release preference of rain water is underestimated by $\overleftarrow{p_{Q0}}(t,\tau)$, and let $\overleftarrow{p_{Q0}}(t,\tau) = (1 - \Omega_{Q_0}(t, S_T(t,\tau)) \cdot f(t,\tau)$ according to Eqs (15)-(16);

(3) Update the improved cumulative age-ranked SAS function $\Omega_Q(t, S_T(t,\tau))$ based on $\overleftarrow{p_{Q0}}(t,\tau)$ obtained at step (2).

In the study of lake Taihu, constraint 2 is not satisfied only when $\tau$ is small. This is because the water parcels of small age in outflow rivers are mostly rain water parcels which enter at the lake surface near the outlets and have shorter path lines comparing to the path lines of river water parcels. Therefore, the value of $\overleftarrow{p_{Q0}}(t,\tau)$ is set to be $(1 - \Omega_{Q_0}(t, S_T(t,\tau)) \cdot f(t,\tau)$ when constraint 2 is not satisfied, indicating that the water parcels of age $\tau$ in outflow rivers are dominated by rain water

parcels.

In this way, the volume of rain water of different ages in the lake can be partitioned from $S_T(t,\tau)$ using the rainfall mixing model. Fig. 3 (b) illustrates the evolution of the volumes of rain and river water by coupling the age master equation model and rainfall mixing model. However, in the incorporated model, the two shape parameters $(\alpha, \beta)$ still remain to be determined, so the tracer transport model is introduced to determine the two shape parameters and calibrate the model.

**3.2.4    Comparison between the incorporated model and the previous TTD model in other lakes**

As is introduced in section 1, the TTD model developed by Smith et al. (2018) evaluates the TTD, RTD and SAS function of the mixed lake water without differentiating rain water and river water. Then, Smith et al. (2018) classified several lake mixing patterns in both the vertical and horizontal directions. They identified two vertical mixing patterns: (1) uniform mixing with depth over the entire lake and equal selection from different depths; (2) young water near the surface (stratified lake)

and outflow dominated by the top layer. In the horizontal direction, they classified four mixing patterns: (1) uniform mixing for discharge; (2) rapid through flow of surface water; (3) incomplete mixing for discharge; (4) more complex mixing across the lake. As a result, the study (Smith et al., 2018) summarized a total of eight mixing patterns based on the combination of these vertical and horizontal mixing patterns. For Lake Taihu, the incorporated model describes one of these classified mixing patterns, characterized by uniform mixing with depth and complex mixing horizontally.

Moreover, the incorporated model considers not only the mixing of aged lake water in the vertical and horizontal directions but also the mixing of rain water and river water in the lake. The rainfall mixing model separates the TTDs and RTDs of rain water and river water. Theoretically, substantial differences in TTDs and RTDs are expected between rain water and river water in the lake due to the "source zone dispersion" (Fiori et al., 2009), which refers to the increase in TTD variance with the source zone variance. Consequently, the TTD of rain water is expected to have a wider range than that of river water in the lake,

as the entire lake surface can receive rain water. This difference in TTDs is all the information that can be inferred from the previous studies, and the incorporated model will advance the understanding of complex mixing dynamics and storage release preference of the lake.



### 3.3   Isotope mass balance model with multiple sources

To evaluate time-variant TTD of Lake Taihu, the tracer transport model, i.e., the isotope mass balance model is employed.
The stable isotope deuterium ($^2$H) is selected as the tracer to resolve how the water parcels of different ages are mixed and transported in the lake. The transport of deuterium in the lake is mainly affected by mixing and evaporative fractionation.

The isotopic composition was measured both in the lake and in the rivers shown in Fig. 2. As the TTD in each outflow river is not estimated in this study, the isotope composition in lake storage rather than in outflow rivers is to be calculated by the isotope mass balance model.

According to the definition of RTD, the mean deuterium concentration in lake water is calculated by integrating the contribution of water of different ages:

$$c_{mean}(t) = \int_{-\infty}^{t} c_L(t, t - t_{in}) \cdot p_s(t, t - t_{in}) \mathrm{d}t_{in} \tag{17}$$

where $t_{in}$ is the input time; $t$ is the clock time; the residence time is $\tau = t - t_{in}$; $c_L(t, t - t_{in})$ is the deuterium concentration in the lake water of age $t - t_{in}$; $p_s(t, t - t_{in})$ is the RTD of lake water.

In Eq. (17), the calculation of $c_{mean}(t)$ requires the determination of $c_L(t, \tau)$. However, in previous studies (Benettin et al., 2017; Queloz et al., 2015), the model to calculate $c_L(t, \tau)$ only includes one source of water, i.e., the rainfall; and the governing equation to determine $c_L(t, \tau)$ is:

$$\frac{\mathrm{d}\left(s\left(\tau + t_{in}, \tau\right) \cdot c_L\left(\tau + t_{in}, \tau\right)\right)}{\mathrm{d}\tau} = - c_L\left(\tau + t_{in}, \tau\right) \cdot F_{\text{out}}\left(\tau + t_{in}\right) \cdot \overleftarrow{p_Q}\left(\tau + t_{in}, \tau\right)$$
$$- \epsilon \cdot c_L\left(\tau + t_{in}, \tau\right) \cdot E\left(\tau + t_{in}\right) \cdot \overleftarrow{p_E}\left(\tau + t_{in}, \tau\right) \tag{18}$$

where $s\left(\tau + t_{in}, \tau\right)$ is the volume of water of age $\tau$ in storage; $\overleftarrow{p_Q}\left(\tau + t_{in}, \tau\right)$ and $\overleftarrow{p_E}\left(\tau + t_{in}, \tau\right)$ are the backward TTD in
river outflow and evaporation, respectively; $\epsilon$ is the evaporative fractionation factor which is defined as the ratio of deuterium concentration in vapour to deuterium concentration in lake water (Gat, 1996). $\epsilon$ is assumed to be a constant for simplicity and will be determined by model calibration later. The solution of Eq. (18) is:

$$c_L(t, \tau) = c_{in}(t - \tau) \cdot e^{(1-\epsilon)\int_0^{t - t_{in}} \frac{E(t)\overleftarrow{p_E}(t, \tau)}{S(t)p_s(t, \tau)} \mathrm{d}\tau} \tag{19}$$

where $c_{in}(t, \tau)$ is the deuterium concentration in the input water parcels at input time $t - \tau$. In the isotope balance model of
Lake Taihu, the lake water comes from both rainfall input and river inflow, so the deuterium concentration $c_L(t, \tau)$ will be calculated as follows:

$$c_L(t, \tau) = \frac{s_{rain}(t, \tau)c_{L\_rain}(t, \tau) + s_{river}(t, \tau)c_{L\_river}(t, \tau)}{s(t, \tau)} \tag{20}$$

where $c_{L\_rain}(t, \tau)$ and $c_{L\_river}(t, \tau)$ are the deuterium concentration of rain water and river water of age $\tau$ in the lake respectively. $c_{L\_rain}(t, \tau)$ and $c_{L\_river}(t, \tau)$ are calculated by Eq. (19), given the initial deuterium concentrations in rain and
river water, $c_{in\_rain}(t, \tau)$ and $c_{in\_river}(t, \tau)$. The derivation of Eq. (19) from Eq. (18) is referred from the study of Queloz et al. (2015).





It is important to note that the calculated $c_L(t, \tau)$ may be biased, if the sources of water are not differentiated. This difference arises because, in the TTD model without distinguishing between rain and river water, the only method to calculate $c_L(t, \tau)$ is by utilizing the mean deuterium concentration of all input sources. This mean concentration assumes that rain and river water are well mixed before entering the lake. However, in many cases, such as in Lake Taihu, the source waters are not well mixed within the system, resulting in a different ratio of the volumes of rain water to river water in the lake comparing to the ratio in the source waters. A detailed explanation, including equation derivation, of this biased outcome is provided in SI (Text S8).

## 4 Model implementation

As is introduced in subsection 3.2 and 3.3, the age master equation model and the isotope mass balance model with multiple sources are coupled by the residence time of rain and river water in the lake. The unknown shape parameters of the age-ranked SAS function can be estimated by calibrating the age master equation model and isotope mass balance model using the monitored isotope data.

Before model implementation, the mean isotope content $\delta^2\mathrm{H}_{\mathrm{mean}}(t)$ of the whole lake and the isotope content in precipitation and inflow rivers are calculated first. The monitored isotope data are presented in SI, the corresponding sampling points are shown in Fig. 2. Overall, the seasonal variation of isotope content at these sampling points is similar, that is, the isotope content reaches the maximum value in May 2013 and June 2014 and the minimum value in Feb 2014, but the amplitude of the variation may be different among these points.

To evaluate the mean isotope content $\delta^2\mathrm{H}_{\mathrm{mean}}(t)$ in Lake Taihu, Lake Taihu is divided into several sub-regions as shown in Fig. 5. The boundary of each sub-region is determined by the two adjacent sampling points, ensuring equal distances from the boundary to the two adjacent points. The isotope content in each sub-region is assumed to be uniform and equal to the isotope content at its respective sampling point. Given the lake's shallow nature, the water volume of these sub-regions is assumed to be proportional to their surface area. In this way, the mean isotope content in the lake $\delta^2\mathrm{H}_{\mathrm{mean}}(t)$ is evaluated by the weighted average of isotope content in each sub-region. Meanwhile, except one daily sampling stations, the other sampling points are at seasonally sampling frequency, so the discrete Fourier transform analysis is applied to rebuild the monthly isotope content at all the sampling points by combing the seasonal and daily sampled data. The implementation of the discrete Fourier transform analysis is presented in SI. As the isotope content in lake water is rebuilt at a monthly frequency, the time step of model implementation is one month.

Moreover, the observed isotope contents in rivers connected to Lake Taihu are presented in SI. Generally, the isotope compositions in rivers in sub-basins ① and ② are smaller than the isotope content in rivers located at the other sub-basins. Sub-basins ① and ② also contribute the most runoff to Lake Taihu. To calculate the mean isotope content in all inflow rivers, the mean isotope content in inflow rivers at each sub-basin is estimated first. Additionally, this calculation accommodates situations where the water inflow rates of some rivers are not available, but the total inflow rates at all sub-basins are known (see SI). The





mean isotope content in inflow rivers in each sub-basin is evaluated as follow:

$$\delta^2 H_{basin} = \sum_i w_i \delta^2 H_i + w_j \frac{1}{n} \sum_k \delta^2 H_k \tag{21}$$

where $n$ is the number of rivers with unknown inflow rates; $\delta^2 H_i$ and $\delta^2 H_k$ represent the isotope contents in $i^{th}$ inflow river

with known inflow rate, and $k^{th}$ inflow river with unknown inflow rate. $w_i$ is the fraction of flow rate of $i^{th}$ inflow river, i.e.,

$w_i = \frac{f_i}{F_{basin}}$ (where $f_i$ is the inflow rate of $i^{th}$ river, $F_{basin}$ is the inflow rate of a particular sub-basin); and $w_j$ is the fraction of

total flow rate in all the inflow rivers with unknown flow rate, i.e.; $w_j = 1 - \sum_i w_i$.

Meanwhile, the isotope content in rivers is monitored seasonally. A cubic spline interpolation method is applied to obtain the

monthly isotope content data. Since the deuterium isotope in river water always exhibits high value in summer and low value

in winter, the seasonal deuterium isotope distribution is adequate to reflect the overall monthly fluctuation throughout the year.

Moreover, as the time step in the model is one month, the influence of fluctuation of isotope content at the timescale smaller

than one month will vanish.

Fig. 6 presents the calculated mean monthly isotope content in the lake, inflow rivers and precipitation based on the two-

year observed isotope data. These calculated isotope data will be further applied to calibrate the age master equation model

and isotope mass balance model. In this study, shape parameters of age-ranked SAS function and evaporative fractionation

factor are the model parameters which are unknown and to be determined by the model calibration. These model parameters

determine the time-variant TTD and RTD of Lake Taihu. Therefore, the aim of the model calibration is to identify the model

parameters which best fit the output of the coupled model, i.e., the mean deuterium isotope content in the lake. The calibration

problem is a typical inverse problem.

The inverse problem is solved by sequential Monte Carlo (SMC) method with a Markov Chain Monte Carlo kernel using the

python package PyMC3 Salvatier et al. (2016). Before implementing SMC method, prior uniform distributions are assigned

to the shape parameters and evaporative fractionation factor. Comparing with the shape parameters introduced in Eq. (10), the

parameters related to age-ranked SAS function are scaled as follow for a better implementation of sampling process in SMC

method:

$$\alpha = \frac{1}{10} e^{\frac{a}{2}}; \quad \beta = \frac{1}{10} e^{\frac{b}{2} + \frac{k}{2} \cdot w(t)} \tag{22}$$

The bounds of the prior uniform distribution (see Table 1) are derived from the prior knowledge of the model parameters

before solving the problem. In this study, they are obtained by several trial runs of the model.

**Table 1.** Bounds of prior uniform distribution of model parameters

| Parameters | $a$ in $\Omega_Q$ | $b$ in $\Omega_Q$ | $k$ in $\Omega_Q$ | fractionation factor $\epsilon$ |
|---|---|---|---|---|
| Lower bound | 0 | 0 | -5 | 0.8 |
| Upper bound | 15 | 15 | 10 | 1 |

During the implementation of SMC method, the age master equation models are solved numerically with a Euler-Forward

scheme with the one-month time step, which is modified from the algorithm by Benettin and Bertuzzo (2018). The numerical




routines of SMC method and Euler-Forward scheme is presented in SI. Posterior distributions of the model parameters are
the results of SMC method. The time-variant TTD, RTD and age-ranked SAS function can be further evaluated based on the
posterior distributions.

## 5 Results and discussion

### 5.1 Calibration result

The marginal distributions of posterior distributions of the model parameters calculated by SMC method are presented in Fig.
7. A higher value of the marginal distribution indicates smaller errors of the calibrated results. The parameters $a$ and $b$ fall
into the ranges of (5.87, 14.99) and (2.94, 12.48) respectively. The ranges are determined by the 95% highest posterior density
interval of model parameters. The parameter $k$ falls into the range of (-2.3315, 9.6191); and the evaporative fractionation factor
$\epsilon$ falls into the range of (0.9367, 0.9739). The implication of these model parameters will be discussed in the following sections.

The simulated mean isotope content in Lake Taihu calculated by the parameters from posterior distribution is shown in Fig.
8. During the calculation, 100 sets of model parameters are drawn randomly from the posterior distributions to illustrate the
goodness of fit of the results of SMC method. The best set of model parameters is: $a = 12.05$, $b = 7.58$, $k = 3.3956$, $\epsilon = 0.9506$.
In Fig. 8, it can be seen that the improved age master equation model and isotope mass balance model with multiple sources
reproduce the monthly variation of deuterium isotope content in the lake very well. The 50% and 95% posterior predictive
probabilities exhibit narrow bounds, with almost all the observed data falling within the bound of 95% posterior predictive
probability.

### 5.2 Time-variant TTD and RTD of river water, rain water and all water

Given the posterior distribution of model parameters, the time-variant TTD and RTD of all water, river water and rain water
are evaluated. Prior to discussing the characteristics of TTD and RTD, it should be noted that the travel time shown in Figs.
9 (a1)-(a3) is the ages of water parcels in outflow rivers rather than in vapour. This is because, based on the assumption of
no selection preference of lake water by evaporation, the TTD of water in vapor always equals the RTD of lake water, so the
TTD in vapour is not discussed. For simplicity, the travel time and TTD in the following text represents the travel time and its
distribution of water in outflow rivers.

#### 5.2.1 Characteristics of TTD and RTD of all water

In Fig. 9 (a3), the travel time at the peak of the TTD of all water is negatively correlated with precipitation (Fig. 9 (a4)). As the
lake volume increases with precipitation (shown in SI), the travel time at the peak of the TTD of all water is also negatively
correlated with the lake volume. Besides, if looking at the fluctuation of travel time at a particular time, the travel time of most
water parcels in outflow rivers falls into a narrow range with length between two to three months as shown in Fig. 9 (a3). For





example, in November 2013, 82.8% of water parcels had the travel time ranging between 4 and 6 months, while 14.7% of water parcels had the travel time smaller than 4 months. The physical control factors of this characteristic are discussed in the next subsection.

As for the RTD of all water in the lake (Fig. 9 (b3)), it can be seen that most water parcels in Lake Taihu have the residence time less than of 9 months. Meanwhile, the influence of each input event can be tracked along the diagonal direction of Fig. 9 (b3). For example, the input event occurred in October 2013 has a large influence on RTD in the following months until February 2014 (indicated by the white oblique line in Fig. 9 (b3)).

Based on the TTD and RTD of all water (Figs. 9 (a3) and (b3)), the mean travel time (MTT) of all water in outflow rivers and the mean residence time (MRT) of all water in Lake Taihu are calculated and presented in Fig. 10. The bounds of 50% and 90% posterior predictive probabilities in Fig. 10 are calculated from 1000 randomly selected sets of model parameters according to the posterior distribution (Fig. 7). In Fig. 10, the MTT ranges from 4 to 8 months during the two years, and the MRT ranges from 3 to 5 months. Both MTT and MRT decrease in October 2013 and August 2014 when the precipitation is high. In addition, the MTT in outflow rivers is always larger than MRT in the lake. This implies that the preferential flow of young water in Lake Taihu is not significant and the flow of water in the lake is more like piston flow, so that the older water parcels may have greater chances leaving the system than younger water parcels. The flow preference in Lake Taihu will be discussed in details in section 5.3.

### 5.2.2 Characteristics of TTDs and RTDs of rain water and river water and their implications

As discussed previously, the calculated TTD of all water in outflow rivers shows that about 80% of water parcels have similar travel time falling into a narrow interval of the length of 2-3 months. However, over 10% water parcels still have the travel time which is smaller than the travel time in the narrow interval, so several questions arise: are the water parcels with smaller travel time from rainfall? What are the differences between the TTD and RTD of rain water and the TTD and RTD of river water? How do the transport of rain water and river water in the lake shape the TTD and RTD of all water? These issues can be well understood by the results of the rainfall-tracking travel time distribution model, which encompass the TTDs and RTDs of rain water and river water.

The calculated TTDs of rain water and river water are shown in Figs. 9 (a1)-(a2). The differences between them are:

(1) The TTD of river water (Fig. 9 (a2)) has similar pattern as the TTD of all water (Fig. 9 (a3)), with a narrow interval of 2-3 months at a particular time; in contrast, the TTD of rain water (Fig. 9 (a1)) spreads in a wider range (starting from zero age) than the TTD of river water (Fig. 9 (a2)). This difference is mainly caused by the "source zone dispersion" Fiori et al. (2009).

(2) The variation of TTD of rain water (along the oblique line in Fig. 9 (a1)) is linked to the variation of rainfall intensity, for example, when the rainfall intensity is high, the volume of rain water from this rainfall event is high in outflow rivers in the subsequent months. However, the variation of TTD of river water (along the oblique line in Fig. 9 (a2)) is not correlated with the rainfall intensity and river input volume.

Referring to the rainfall mixing model shown in Fig. 4, the first difference comes from the distinct source zones, i.e., the entry points between rain and river water parcels. The entry points of rain water spread out over the lake surface, while the





entry points of river water are located at the lake shore connected to the inflow rivers. The different source zones account for the board range of the TTD of rain water compared to that of the TTD of river water. This phenomenon is known as the "source zone dispersion", which is introduced in section 3.2.4.

The second difference is mainly caused by the distinct spatial distributions of rain and river water parcels in the lake. Specifically, the rain water parcels of the same age (i.e., from the same rainfall event) spread out along the lake surface and flow to the outflow rivers gradually, but the river water parcels of the same age remain concentrated in a narrow band within the lake as indicated by the rainfall mixing model in Fig. 4. This will be more clear after the discussion of decrease rates of RTDs of rain water, river water in the Lake in the following paragraph of this section.

The calculated RTDs of rain water, river water and all water are shown in Figs. 9 (b1)-(b3). Comparing Figs. 9 (b1) and (b2), the shapes of the RTDs of rain and river water are similar. However, the decrease rates of the volumes of rain and river water parcels from the same rainfall event or river input time are different. This difference is shown in Fig. 11, where the volumes of rain water and river water from the same rainfall event or river input time in Lake Taihu are tracked. Specifically, the decrease rates of rain water in the lake are approximately constants in all the sub-figures in Fig. 11, but the decrease rates of river water increase suddenly after a certain age. Referring to the rainfall mixing model, the sudden decrease of the volume of river water in the lake is attributed to the particular spatial distribution of river water within the lake, where most of the river water of the same entering time is distributed in a narrow band and flows out of the lake within a narrow interval of length of 2-3 months. The difference in the decrease rates of the volumes of river water and rain water in the lake is the direct reason leading to the second difference between the TTDs of river water and rain water.

### 5.2.3 How do the river and rainfall inputs shape the TTD and RTD of all water in the lake?

Figs. 9 (a3) and (b3) present the TTD and RTD of all water respectively. Theoretically, the TTD of all water is the weighted average of the TTDs of river water and rain water, and the weights are defined as the fractions of the volumes of river water and rain water in outflow rivers respectively. The same is true for the RTD of all water, but the weights for RTD are the fractions of the volumes of river water and rain water in thr lake. Fig. 12 shows the calculated weights of TTD of rain water (Fig. 12 (a1)) and the weights of RTD of rain water (Fig. 12 (b1)) in Lake Taihu, the two weights range from 0.15 to 0.25. Therefore, the shapes of TTD and RTD of all water are mostly determined by the shapes of TTD and RTD of river water. Meanwhile, based on the weights and shapes of TTDs of rain and river water, it can be inferred that the water with longer travel time in outflow rivers is dominated by river water with a larger proportion, while the water with shorter travel time is dominated by rain water with a smaller proportion.

Moreover, if comparing the weights to the ratio of the rainfall rate to the total inflow rate in Fig. 12, it can be found that the weights are mostly determined by this rainfall input ratio. Therefore, it can be deduced that in other lakes with high rainfall input ratios, the characteristics of TTDs of rain water will become more significant in the characteristics of TTDs of all water. Generally, the rainfall input ratio varies in different climate conditions, so the characteristics of TTD and RTD of all water varies accordingly.



## 5.3 Time-variant SAS function

### 5.3.1 Characteristics of SAS functions of all water

SAS function describes how the water parcels are released from lake storage. The age-ranked SAS function links TTD and age-ranked storage volume in the age master equation model. In this study, three model parameters $a$, $b$ and $k$ are used to
quantify the cumulative age-ranked SAS function. As shown in Fig. 7, the best fitted shape parameters of cumulative age-ranked SAS function follow the relation: $\alpha > \beta > 1$. This relation results in the peak of SAS function appears in the old water parcels, indicating that Lake Taihu prefers releasing older water in outflow rivers. Meanwhile, the parameter $k$ is larger than zero, which means that the shape parameter $\beta$ increases with lake volume, that is, more young water is released when lake volume is high.

Figs. 9 (c1)-(c3) presents the absolute SAS functions of rain water, river water and all water in Lake Taihu. The absolute SAS function is defined as the ratio of backward TTD to RTD Botter et al. (2011): $aSAS = \frac{TTD}{RTD}$. Generally, the probability of water leaving the system increases with the value of the absolute SAS function. In Fig. 9 (c3), the age at the peak of the absolute SAS function of all water is negatively correlated with the lake storage volume due to the positive $k$ value.

In Fig. 9 (c3), the SAS function of all water in Lake Taihu indicates that old lake water with the age of 5-9 months has higher
possibility leaving the system than young water with the age of 1-4 months; and when lake volume is high (June 2013-October 2013 and July 2014-September 2014), the age of the lake water with higher possibility leaving the lake decreases to 4-6 months, which is known as the inverse storage effect.

### 5.3.2 Characteristics of the SAS function of rain water and river water

In Figs. 9 (c1) and (c2), the calculated absolute SAS functions of rain water and river water indicate that the lake prefers
releasing old rain water and river water from the lake to outflow rivers, but the young rain water has higher possibility leaving the lake than the young river water. This is obvious as some entry points of rain water are closer to the outflow rivers, comparing to the entry points of river water.

As for the release preference of river water, the volume of river water of different ages in outflow rivers is mostly determined by the path lines flowing through the lake. The SAS function of river water indicates that the preferential flow in the lake is not
significant, and the old river water is preferred to be released from the lake through outflow rivers.

Fig. 9 (c3) presents the calculated absolute SAS function of all water. Similar to the TTD and RTD of all water, the absolute SAS function of all water is the weighted average of the absolute SAS functions of rain water and river water:

$$aSAS_{\text{all}} = aSAS_{\text{river}} \cdot \frac{RTD_{\text{river}}}{RTD_{\text{all}}} \cdot \frac{F_{out\_river}}{F_{out}} + aSAS_{\text{rain}} \cdot \frac{RTD_{\text{rain}}}{RTD_{\text{all}}} \cdot \frac{F_{out\_rain}}{F_{out}} \tag{23}$$

where $\frac{F_{out\_rain}}{F_{out}}$ and $\frac{F_{out\_river}}{F_{out}}$ are the fractions of the volumes of rain and river water in outflow rivers; $\frac{RTD_{\text{river}}}{RTD_{\text{all}}} \cdot \frac{F_{out\_river}}{F_{out}}$ and
$\frac{RTD_{\text{rain}}}{RTD_{\text{all}}} \cdot \frac{F_{out\_rain}}{F_{out}}$ are the weights of absolute SAS function of river water and rain water respectively. The derivation of Eq. (23) is based on the definition of absolution SAS function (see SI, text S9). In Lake Taihu, the difference between $\frac{RTD_{\text{river}}}{RTD_{\text{all}}}$ and





$\frac{RTD_{\text{rain}}}{RTD_{\text{all}}}$ is quite small, so the characteristics of absolute SAS function of all water is dominated by the fractions of the volumes of rain water $\frac{F_{out\_rain}}{F_{out}}$ and river water $\frac{F_{out\_river}}{F_{out}}$ in outflow rivers.

## 5.4 Implication of the isotope data

The parameters in the age master equation model are calibrated based on the isotope data in Lake Taihu, but how the model parameters determine the mean isotope content in the lake is unclear yet. A sensitivity analysis of the model parameters is crucial to increase the reliability of the model. Moreover, the two-year seasonal variation of the spatial distributions of isotope data is a valuable data set to investigate the relations among the calculated TTD, RTD and the spatial distribution of the isotope data.

### 5.4.1 Determination of model parameters on the mean isotope content in the lake

To explore the determination of the model parameters on the mean isotope content in the lake, a series of sensitivity analysis on the model parameters are conducted. In Fig. 13, the model parameters $a$ and $b$ determine the skewness of the beta distribution, i.e., the release preference of young water or old water from the lake to outflow rivers. Specifically, the release preference of old water increases with $a$, and the release preference of young water increases with $b$. Since the deuterium isotope is more 575 concentrated in old water, the value of $a$ and $b$ will certainly influence the mean deuterium concentration in the lake. In Fig. 13 (a), when $a$ increases (i.e., the volume of old water leaving the lake increases), the amplitude of the fluctuation of the mean isotope content in the lake decreases; and when $a$ decreases (i.e., the volume of old water in the lake increases), the mean isotope content in the lake increases. Similarly, in Fig. 13 (b), the determination of parameter $b$ on the mean isotope content in the lake is the opposite of the determination of $a$. As for the parameter $k$ shown in Fig. 13 (c), when $k < 0$, which 580 is identified as the direct storage effect, the amplitude of the fluctuation of mean isotope content in the lake decreases, as the release preference of old water increases when lake volume is large; when $k > 0$, which is identified as the inverse storage effect, the mean isotope content in the lake increases with $k$, and this is caused by the increase of the release preference of young water when lake volume is large. Finally, the influence of parameter $\epsilon$, which is the fractionation ratio of deuterium isotope in the lake, is obvious: the mean isotope content in the lake increases as $\epsilon$ decreases.

Generally, the controls of $a$, $b$ and $k$ on the mean isotope content in Lake Taihu can be summarized as: 1) the amplitude of the fluctuation of mean isotope content in the lake decreases as the release preference of old water increases; 2) the mean isotope content increases as the release preference of young water increases. These determinations is attributed to the higher isotope content in old lake water compared to that in young lake water. Meanwhile, this sensitivity analysis verifies the accuracy of TTDs, RTDs and SAS functions calculated using the model parameters $a$, $b$, $k$ and $\epsilon$. This is because through sensitivity 590 analysis, it has been found that different parameter combinations can yield various patterns of isotope fluctuations, one of which corresponds to the fluctuation pattern observed in Lake Taihu. Therefore, by utilizing the parameter combination that best fits the isotope data, the TTDs, RTDs and SAS functions in Lake Taihu can be accurately calculated.





### 5.4.2  Implication of the spatial distributions of deuterium isotope data

The calculated TTDs, RTDs and SAS functions are lumped functions, while the spatial distributions of deuterium isotope
content (shown in Fig. 14) serve as a "key" to open the black box and reveal the mechanisms underlying the temporal variations
of these functions, providing insights into transport and mixing dynamics within the lake.

Figs. 14 (d)-(l) present the seasonal variations of deuterium isotope content from November 2012 to November 2014. Generally, the deuterium isotope becomes enriched from northwest to southeast. The previous study Xiao et al. (2016) pointed out
that the spatial patterns of the deuterium isotope in the lake are not caused by the changes in local evaporation rate, because
the spatial variations of water surface temperature and local evaporation rate are negligible. Instead, as depicted in the isotope
mass balance model, it is the residence time of lake water that leads to the progressive enrichment of deuterium isotope from
northwest to southeast due to the evaporation fractionation. Therefore, the spatial distribution of deuterium isotope content is
closely related to the water flow direction in the lake.

To provide a clearer depiction of the shift in flow direction, three zones are divided in Lake Taihu based on the spatial
distribution of isotope content and the locations of inflow and outflow rivers: the inflow zone, transition zone and outflow zone
(Fig. 14 (c)). The water flow direction is generally from the inflow zone to outflow zone. In addition, the rivers connected to
the inflow zone and outflow zone are all inflow rivers and outflow rivers respectively. However, there are three rivers connected
to the transition zone with reversible flow directions. Among the reversible rivers, the Wangyu River (Fig. 2), which is located
at the northeast of Lake Taihu and is connected to the Yangtze River, has the largest flow rate. When the lake level is low, the
water is diverted from the Yangtze River to the lake to raise the lake level and dilute the pollution in the lake, so the Wangyu
River is an inflow river. Conversely, when the lake level is high, the Wangyu River becomes an outflow river for flood discharge.
As a result, the flow direction in the lake near the Wangyu River is reversible in time.

Due to the existence of three reversible rivers, the flow direction in the transition zone varies in time, which further influences
the flow paths between the inflow zone and outflow zone. From Figs. 14 (d)-(l), it can be found that the water flow paths in the
north of the lake are highly influenced by the flow direction and flow rate of the Wangyu River.

According to the TTD theory, variations in flow paths play a significant role in determining the temporal variation of SAS
function. Comparing the SAS function in Fig. 9 (c3) with the spatial distribution of isotope data in Fig. 14, it's evident that when
the Wangyu River becomes an outflow river from May 2014 to August 2014 (excluding June 2014), during periods of high
lake volume, there is a preference for releasing younger water from storage, known as the inverse storage effect. Conversely,
in other months of 2014 when the lake volume is low and the Wangyu River serves as an inflow river, the SAS function shifts
towards older water. Below are detailed explanations of the influence of Wangyu River on water flow paths and SAS function
in three typical months.

In February 2014 (Fig. 14 (i)), the Wangyu River was an inflow river, so the water at the inflow zone had to travel through
the entire lake and exit at the outlets of the outflow zone, predominantly flowing from northwest to southeast. Moreover, the
water entered from the Wangyu River at the northeast corner can obstruct the flow from northwest to southeast, potentially
prolonging the travel time from inflow to outflow zones.





In June 2014 (Fig. 14 (j)), there was no water flowing through Wangyu River due to the closed sluice. Flow patterns show a transition between February 2014 (Fig. 14 (i)) and August 2014 (Fig. 14 (k)). Another difference is the spatial gradient of deuterium isotopes, indicating higher velocities and shorter travel times from inflow to outflow zones. Therefore, the storage
release preference shifted slightly towards younger water in June 2014 compared to February 2014.

In August 2014 (Fig. 14 (k)), the Wangyu River became an outflow river due to the high lake level. This allowed water from the inflow zone to exit via rivers in both the transition and outflow zones, creating shorter flow paths compared to February 2014. Additionally, the smaller spatial gradient of deuterium concentration indicated shorter travel times for water flow paths in the lake. Consequently, both the TTD (Fig. 9 (a2)) and SAS function of river water (Fig. 9 (c2)) shifted toward young water
significantly.

In summary, the flow direction and flow rate of Wangyu River play an import role in the control of water flow path, travel time and storage release preference of the lake in 2014. Meanwhile, the similar spatial patterns of deuterium isotope in 2013 indicate the same control of the flow status of Wangyu River on the temporal variations of TTD, RTD, SAS function of the lake. This control is most significant in 2014 due to the large fluctuation of lake volume and the significant variation of the flow
status of the Wangyu River.

## 6 Conclusions

This study quantifies the time-variant TTD, RTD and SAS function of surface water in Lake Taihu using the rainfall-tracking travel time distribution model. This new TTD model is developed to track the fates of river water and rain water in the lake separately. In this model, the age master equation model is coupled with the rainfall mixing model which describes how the
rain water is mixed with the river water in the lake. Then, the TTDs, RTDs and SAS functions of river water and rain water are differentiated.

The calculated time-variant TTD of all water shows that the travel time of most water parcels (about 80%) in outflow rivers falls into a narrow interval of the length of two or three months. The travel time at the peak of time-variant TTD of all water is negatively correlated with precipitation, i.e., the travel time interval ranges from 2-3 months to 7-8 months in time depending
on the precipitation volume. The time-variant RTD of all water shows that most water parcels in Lake Taihu are younger than 9 months. The time-variant SAS function of all water displays an inverse storage effect, i.e., more young water is released when lake volume increases.

The results of the rainfall-tracking travel time distribution model explain the formation of the inverse storage effect. Firstly, the patterns of the TTDs of rain water and river water are completely different due to their different source zones and transport
processes in the lake. The rain water contributes to 15%-25% of outflow volume, and the age of the rain water is between 0-8 months in outflows; the river water contributes the remaining outflow volume, and the age of the river water in outflows falls into narrow intervals with lengths of 2-3 months in length. As the river water contribute most river outflow, the transport of water from inflow rivers dominates the pattern of TTD of all water in outflow rivers. Secondly, the SAS functions of river and



rain water indicates the release preference of old rain and river water from the lake to outflow rivers, but young rain water has
higher possibility leaving the lake than the young river water.

Further analysis of the seasonal variation of spatial distributions of deuterium isotope illustrates that the storage release
preference of Lake Taihu is mainly controlled by the temporal variations of flow paths and velocities in the lake. Intensively
flow regulated rivers, e.g., Wangyu River connecting Lake Taihu and Yangtze River, have a significance influence on the
temporal variation of water flow paths and travel time in the lake.

This study reveals the determination of water flow path on time-variant TTD, RTD and storage selection preference in a
large shallow circular lake. The transport and mixing process of rain water and river water are studied quantitatively with
the assistance of time-series high-spatial-resolution deuterium isotope data. The results are instructional to understand the
hydrological and ecological problems in Lake Taihu. The methods developed in this study may also be employed to investigate
the transport and mixing process in the other large shallow lakes, and the water circulation pattern of Lake Taihu may be a
reference to other large shallow circular lakes with dense river networks around the world.

*Code and data availability.* Dataset and python codes for this research have been archived and are available at Zenodo: Mao (2023). Such
dataset and codes must be findable and accessible with the DOI link: https://doi.org/10.5281/zenodo.10156422.

*Author contributions.* Rong Mao: Conceptualization, Methodology, Data curation, Formal analysis, Writing - original draft; Xin Luo: Con-
ceptualization, Writing - review & editing; Jiu Jimmy Jiao: Project administration, Writing - review & editing, supervision; Xiaoyan Shi:
Formal analysis, Writing - review & editing; Wei Xiao: data provision.

*Competing interests.* The authors declare no conflict of interest.

*Acknowledgements.* This study was supported by a grant from the Zhejiang Institution of Research and Innovation, The University of Hong
Kong (Seed fund 04003), and the seed fund for basic research for new staff in the University of Hong Kong (No. 201909185058) granted to
L.X. Appreciations are given to Taihu Basin Authority of Ministry of Water Resource for the help in accessing to the online dataset.



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



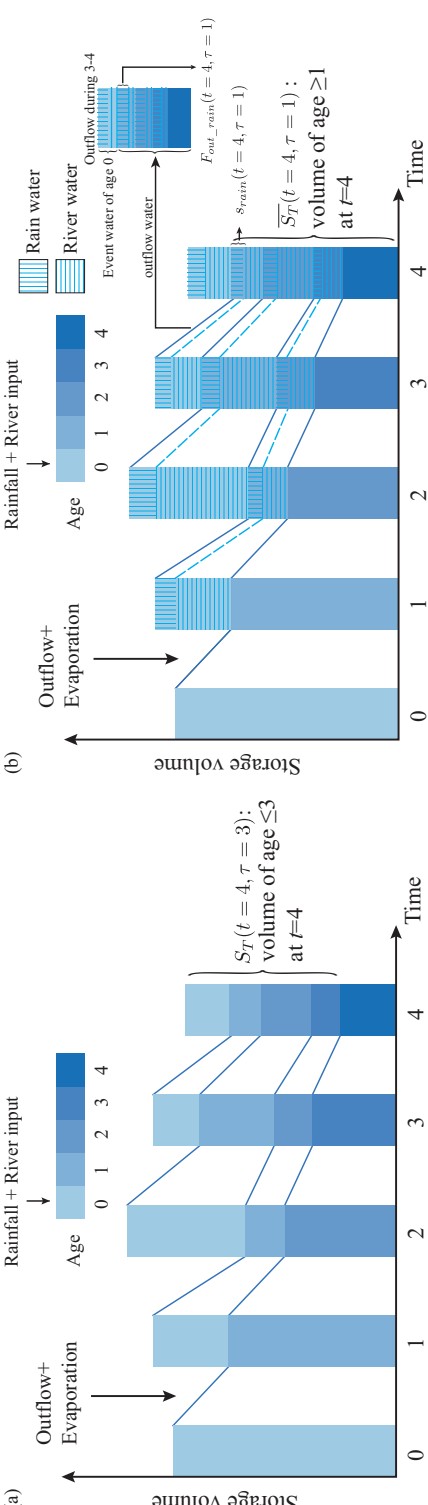

**Figure 3.** (a) Evolution of age-ranked storage $S_T(t,\tau)$ in time; (b) evolution of age distributions of rain and river water in storage in time, and the relation between the volume of aged rain water and the age-ranked storage complement $\overline{S_T}(t,\tau)$, i.e., $s_{rain}(t,\tau) = \overline{S_T}(t,\tau) \cdot f(t,\tau)$.





**Figure 4.** Rainfall mixing model in Lake Taihu. (a) Horizontal mixing and vertical mixing of rain water and river water along flow tube AB; (b) temporal evolution of the age composition along flow tube AB. Vertical structures in (a) and (b) are depicted to demonstrate the age composition along the flow tube. However, in reality, the stratification along the flow tube doesn't exist in Lake Taihu, as rain water is well mixed vertically with the local lake water.





**Figure 5.** Sub-regions of Lake Taihu divided based on the sampling points in the lake.



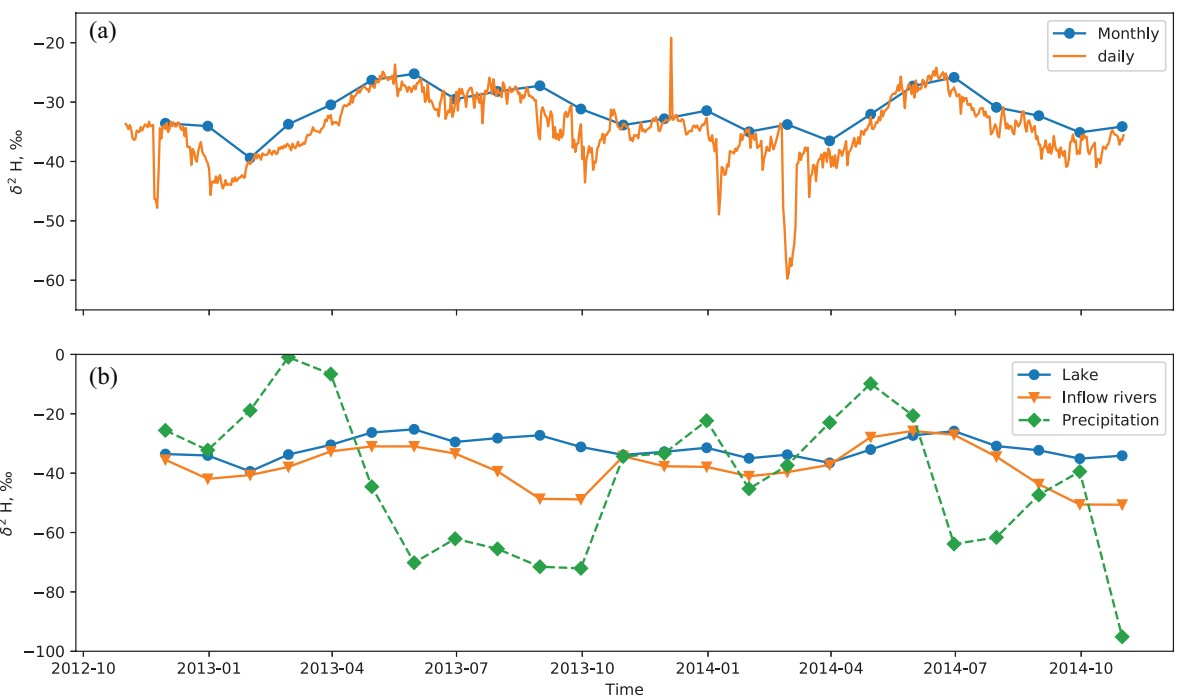

**Figure 6.** Calculated mean isotope content in lake, inflow rivers and precipitation. Sub-figure (a) compares the observed isotope content at the daily sampling point in the lake with the calculated monthly mean isotope content in the whole lake. Sub-figure (b) presents the calculated monthly mean isotope content in the whole lake, inflow rivers and precipitation.




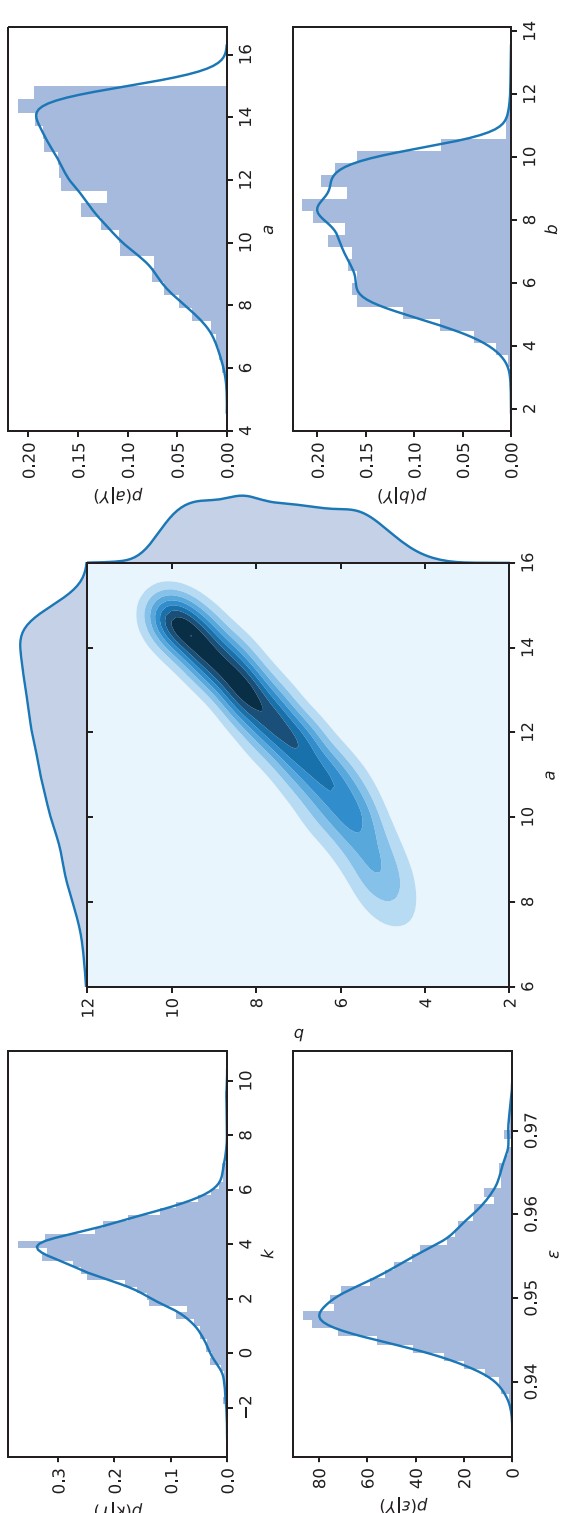

**Figure 7.** Marginal distribution of posterior distribution $p(\Theta|Y)$ for each model parameter. $a$, $b$ and $k$ are scaled parameters, and the fractionation factor $\epsilon$ is not scaled.

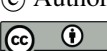



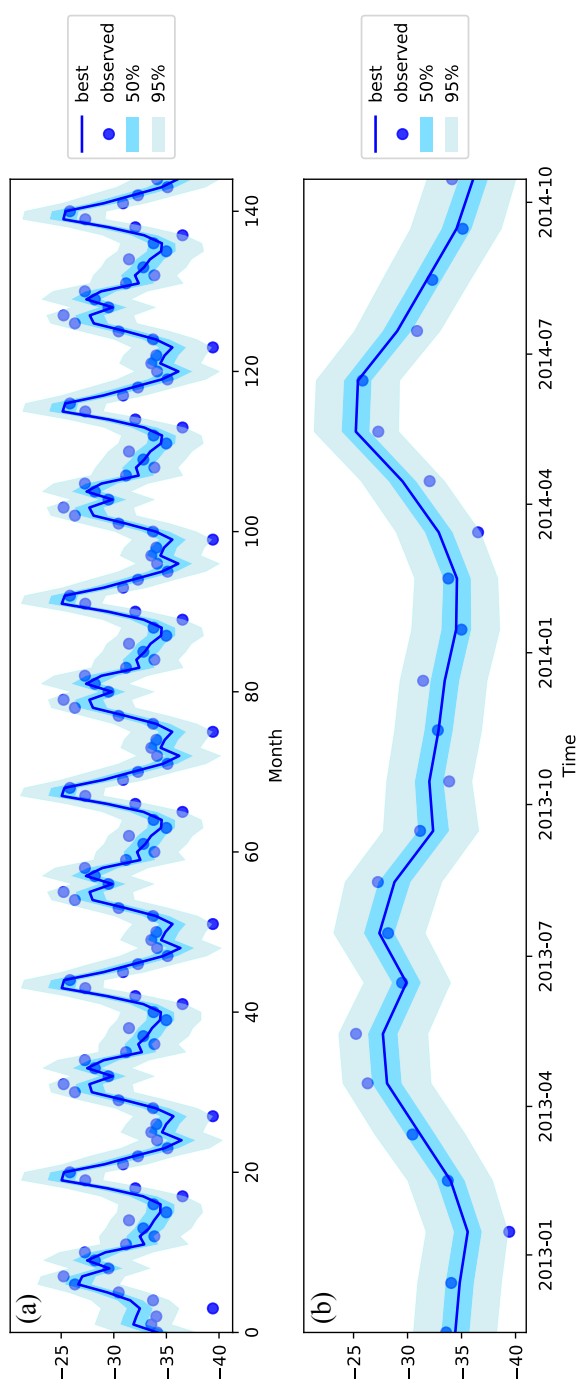

**Figure 8.** Simulated mean isotope content in the lake with 95% and 50% posterior predictive probability bounds is compared with the calculated mean isotope content from observed isotope content in the lake. Sub-figure (a) shows the results in the whole 12 years, and sub-figure (b) shows the results in the last two years of the modeling time.



**Figure 9.** Time-variant TTDs, RTDs and SAS functions of rain water, river water and all water.



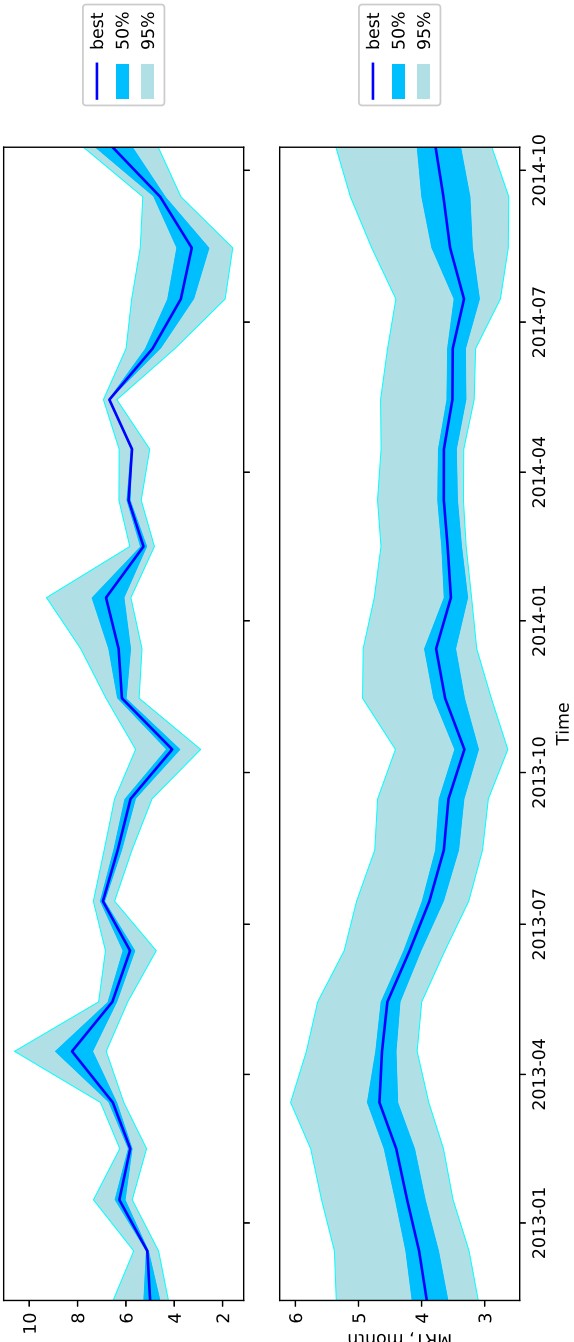

**Figure 10.** Mean travel time and mean residence time of Lake Taihu with 95% and 50% posterior predictive probability bounds calculated by using the posterior distribution of model parameters.



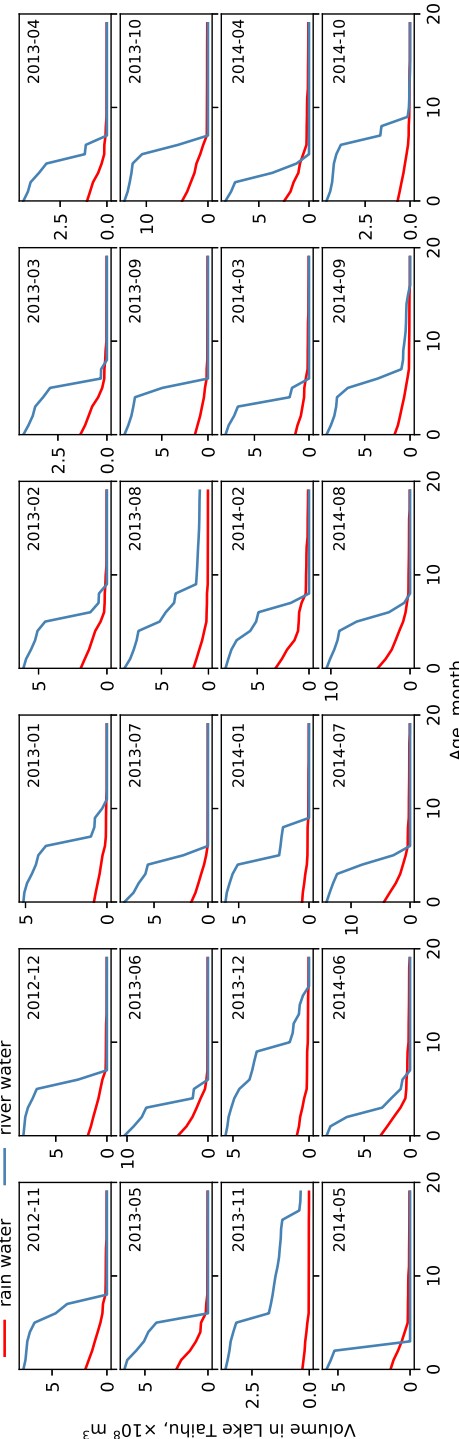

**Figure 11.** Decrease of the volumes of rain water and river water of the same entering time ranging from November 2012 to October 2014 in Lake Taihu.





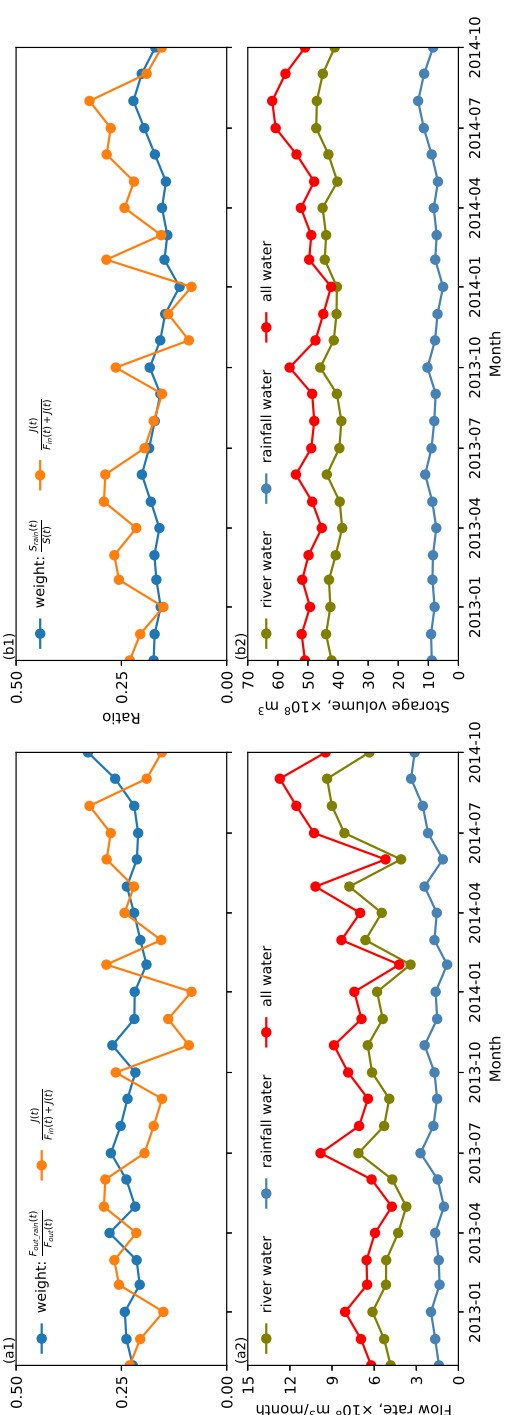

**Figure 12.** Temporal variations of the proportion of rain water in outflow rivers (a1) and in the lake (b1), and the temporal variations of the associated outflow rate of rain water in outflow rivers (a2) and volume in the lake (b2).



**Figure 13.** Sensitivity analysis of model parameters. Best in the figure means the parameters fits the monitored data best, i.e., $a = 12.05, b = 7.58, k = 3.3956, \epsilon = 0.9506$



**Figure 14.** Seasonal variations of the spatial distribution of deuterium isotope content in Lake Taihu ((d)-(l)). (a) Temporal variation of lake volume ($10^8\,\mathrm{m}^3$); (b) temporal variation of flow direction and rate of Wangyu River ($10^8\,\mathrm{m}^3/\mathrm{month}$), and the flow rates are also marked in (d)-(l); (c) three flow zones in Lake Taihu.