# Peer review of "A rainfall-tracking travel time distribution model to quantify mixing and storage release preference in a large shallow lake by two-year stable isotopic data"

_EGUsphere, 2024_

## Author Comment (AC1)

**Reply to the RC1**

In this file, the comments from RC1 are in **black font**; our replies below are in **blue font**.

In my opinion, the manuscript will be a great addition to the journal and be of interest to its readers. Mao et al targets the variations in time and in space of residence time and travel time at Lake Taihu, China, by processing and analyzing measurements of deuterium in the lake. Temporal scales in lakes have been correlated to water quality issues, therefore knowing how long a water parcel remains in the lake depending on the meteorological conditions will provide a clearer picture on how to better manage water quality in Taihu Lake. For example, their results show that depending on the inflow of water (through rivers and rain) the travel time can decrease by half.

Reply: We appreciate your acknowledgment of our manuscript's contribution to the journal. As you summarized, our manuscript aims to quantify the residence time, travel time, mixing, and storage release preferences of Lake Taihu, which are essential for investigating the eutrophication problem in this lake. Below are our point-by-point replies, which we hope will address your comments satisfactorily.

1. There have been publications mentioning Lake Taihu's residence time. I am thinking for example of Xu et al 2009 (https://doi.org/10.4319/lo.2010.55.1.0420), Xu et al 2015 (https://doi.org/10.1021/es503744q) and Paerl et al 2014 (https://doi.org/10.1371/journal.pone.0113123). Interestingly the value of residence time is not consistent among these publications, but they remain below the 1 year mark. I think the manuscript would benefit by including a comparison and a possible explanation of the discrepancy between the publications.

Reply: We accept your valuable suggestions. Including a comparison and an explanation for the discrepancies between the publications is a good way to highlight our contribution. In our current manuscript, we only compare the scientific questions related to residence time addressed by our model with those addressed by Li et al. 2011 (https://doi.org/10.1016/j.ecoleng.2010.11.024) (see lines 74-79 in the manuscript which is being discussed), but, as you noted, there is more literature mentioned or estimated the residence time in Lake Taihu. Therefore, in the revised version, we will expand the literature review in the introduction section and compare our results with the time scales from other studies in the results section.

Here are our comparisons, which will be included in the future revised version:
(1) Firstly, in the literature on the study of Lake Taihu, the terms related to time scales vary widely. Some use "age", some use "retention time", and others use "residence time", each with different definitions, leading to varying numerical values in the literature. We will address the numerical differences caused by these terms in the following comparison.
(2) Secondly, the residence time and travel time of water in Lake Taihu are influenced by various factors, among which wet conditions (or lake water level), water flow path, and water flow velocity are the main factors discussed in our manuscript. These factors can lead to the differences in time scales among literature.

(3) In the study of Li et al. 2011 (https://doi.org/10.1016/j.ecoleng.2010.11.024), the spatial distribution of age in Lake Taihu was simulated. The term "age" in their study represents the locally averaged "residence time", influenced by the local mixing of water parcels with different residence times. This mixing can result from molecular diffusion between adjacent flow lines or the mixing of rainwater into the lake's surface with the local lake water. Consequently, the maximum age in the lake tends to be slightly smaller than the maximum residence time. Li et al. 2011 reported maximum ages of 120-150 days in July and 240-270 days in December. In our study, as shown in Fig. 9(b3) and lines 467-468, the maximum residence time is less than 9 months (varies between 4-5 months to 8-9 months), consistent with the results of Li et al. 2011.

(4) In the studies of Xu et al 2009 (https://doi.org/10.4319/lo.2010.55.1.0420), Xu et al 2015 (https://doi.org/10.1021/es503744q), the term "retention time" is used,which is similar to the definition of travel time, i.e., the time of water spends from inflow to outflow. Theoretically, travel time is not fixed at a specific moment, as some outflow rivers have shorter travel times while others have longer ones. Therefore, we quantify the travel time distribution in our study (Figs. 9 a(1)-a(3)). The temporal variation of mean travel time is calculated and shown in Fig 10. According to our findings, the travel time in outflow rivers varies between 2-4 months (high lake water level) and 7-9 months (low lake water level). In Fig. 10, the mean travel time in outflow rivers ranges from 4 to 8 months. However, retention times reported in the literature vary significantly, such as 284 days (Xu et al., 2009), 180 days (Xu et al., 2015), and 5 months (Qin et al., 2007). These retention times fall within the range of our calculated travel time distribution, which varies in time and space depending on lake water level, water flow paths, water flow velocities, etc.

(5) In the study by Paerl et al. (2014) (https://doi.org/10.1371/journal.pone.0113123), we did not find any information regarding water transport time scales.

Overall, there is currently a lack of unified and systematic understanding of the time scales of water transport in Lake Taihu. Utilizing travel time distribution theory and storage selection functions, we have conducted a comprehensive and systematic study on the time scales of water transport in Lake Taihu. We have quantified the travel time distribution, residence time distribution, mixing, and storage release preferences of lake water over time. This research is essential for future studies on the ecological and water quality issues of Lake Taihu.

Qin, B., Xu, P., Wu, Q., Luo, L., & Zhang, Y. (2007). Environmental issues of lake Taihu, China. Eutrophication of shallow lakes with special reference to Lake Taihu, China, 3-14.

2. There are portions of paragraph in Section 5 that appear contradictory to the results described in the abstract. For example, at the end of Section 5.2.1, lines 477-478, Mao et al states that "older water parcels may have greater chances leaving the system than younger water parcels". However, in the abstract it is stated that "the release preference shifts toward younger water when lake volume is large". A reorganization of the subsections would help to make things clearer.

Reply: We apologize for the confusion caused by the sentences in section 5 and the abstract. We

will revise the abstract sentence as follows: "The release preference shifts toward younger old water when the lake volume is large", which means the release preference is still old water, but the age of these old water becomes a little bit younger when lake volume is large. (see lines 544-546 in the discussed manuscript). This is primarily due to two reasons: (1) the mean travel time and residence time decrease with larger lake volume and flow velocity; (2) there are shorter flow paths in the lake when lake volume is large. We will carefully review all sentences in section 5 to prevent similar issues in the future.

3. In the introduction, lines 51-52, the authors mention that only Smith et al. 2018 applied the travel time distribution theory to a lake. Unless the authors were implicitly focusing on the usage of isotopic compound, I disagree with the authors. Temporal scales have been addressed in previous work in lakes. For example, Rueda and Cohen, 2005 (https://doi.org/10.4319/lo.2005.50.5.1638) looked at variation in space and time of residence time in an embayment of Lake Ontario, Canada.

Reply: We accept this comment. In 51-52, our intention was to say that currently only Smith et al. (2018) have applied travel time distribution theory and storage selection functions to two lakes to quantify the ages and storage release preferences of the lakes. However, in the study by Rueda and Cohen (2005), the TTD theory was at an early stage, and recent development in TTD theory, such as storage selection functions and age master equations put forward in the 2010s, are very helpful to characterize not only RTD but also the mixing and storage release preference of lakes. Moreover, in our study, as described in lines 52-61, we further developed the TTD model to quantify the mixing of rainwater and river water in Lake Taihu.

Therefore, we will revise this sentence as: "To the best knowledge of the authors, while there is study (Rueda and Cohen, 2005) estimating RTD in lakes, there is only one study (Smith et al. 2018) applied the time-variant TTD theory developed in 2010s and storage selection function to quantify the age, mixing and storage release preference of the lake".

4. The authors mention the spatio-temporal variations of the time scales are controlled by horizontal velocities on several occasions (in the abstract and in the conclusion), they did not discuss past work identifying the flow field at Lake Taihu. The manuscript would benefit from including comparisons with past work on average flow field in Lake Taihu Liu et al. 2018 (doi:10.3390/w10060792) shows for example a rather complex average flow field, which would definitely hinder the flow of water from inflow to outflow.

Reply: We accept this suggestion. In section 5.4.2, we will add a comparison of past works in literature on average flow fields as supplementary information to illustrate the implications of spatio-temporal variations in isotope data on the flow field, and to discuss the relationship between the flow field and time scales.

Firstly, the overall water flow direction in Lake Taihu has been reported in many studies, such as Qin et al, 2007, Xiao et al. 2016 (https://doi.org/10.1080/10256016.2016.1147442), indicating a generally flow direction from northwest to southeast. This pattern is generally consistent with the

spatial distribution of isotopes shown in Fig. 14.

As for the temporal variation of flow paths, the isotope data indicate that the water transfer project in Wangyu River plays a significant role in controlling water flow paths. This is confirmed by Liu et al. 2018 (doi:10.3390/w10060792), who investigated the influence of water transfer project on the water circulation patterns. Their study found that when Wangyu River is an inflow river, it significantly obstructs the northwest-to-southeast water flow direction, thereby prolonging the travel time from inflow to outflow zones (as we discussed in lines 624-626). Conversely, in June 2014, when Wangyu River became an outflow river, this obstruction ceased, explaining the shift in storage release preference towards younger old water during that month.

5. I noticed a couple of typos in the text:
Figure 2: There is no sampling point 2. I assume it is supposed to be where sampling point 32 is.

Reply: After examining our dataset, we confirmed that there is indeed no sampling point 2 here. Although the figures are numbered up to 32, as mentioned in line 114, there are only a total of 29 sampling points for isotope data. Isotope data at points 2, 9, and 15 are not available. You can find our isotope data for these 29 sampling points in the supplementary file. Additionally, the original data in xlsx format is available for download from the following DOI link: https://doi.org/10.5281/zenodo.10156422.

Line 304: If I understand properly, s_rain(t,tau) is the volume of rain water aged tau, and not s(t,tau).

Reply: Yes, it should be s_rain(t,tau), we will revise it in the manuscript.

Line 499: please replace "board range" with "broad range".

Reply: Thanks for pointing out this typo, we will correct it in the manuscript.

Line 505: please replace "Lake" with "lake".

Reply: Thanks for pointing out this typo, we will correct it in the manuscript.

---

## Author Comment (AC2)

**Reply to the RC2**

In this file, the comments from RC2 are in **black font**; our replies below are in **blue font**.

Overall reply: After several days of careful evaluation of our model, we are thrilled to inform you that we have derived a new SAS function analytically for water in outflow rivers. The transport and releasing of rain water and river water have been integrated into this new SAS function successfully. The new SAS function can be calculated analytically, without check whether the beta distribution is appropriate at each time step. In this way, the whole calculation process following the same procedures in previous TTD literature. Therefore, this new SAS function makes our model simpler and easier to understand. Below is a brief introduction to our improved model, followed by a point-by-point response to your comments.

In our new SAS function for outflow rivers, we introduce a new variable called the event rain water threshold age (Figure 1). As shown in Figure 1, for water age below this threshold age, only rain water is released from the lake; for water age above this threshold, both rain water and river water can be released. The threshold age can equal 0, which means there is no event rain water in the outflow. The physical interpretation of the event rain water threshold age is that some rainfall enters the lake very close to the outflow rivers, however there is always a distance between the inflow rivers and the outflow rivers.

[Figure]

Figure 1. Illustration of the new SAS function for outflow rivers and the event rain water threshold.

Let $\tau_e$ represent the event rain water threshold age, then the new cumulative SAS function $\Omega_Q(t, S_T(t, \tau))$ is defined as follow:

$$\Omega_Q(t, S_T(t, \tau)) = \begin{cases} \Omega_{Q1} & \text{if } \tau < \tau_e \\ \Omega_{Q2} & \text{if } \tau > \tau_e \end{cases} \quad (1)$$

where $\Omega_{Q2}$ is the traditional cumulative SAS function. In this study, we use the beta distribution to characterize $\Omega_{Q2}$; and $\Omega_{Q1}$ remains to be determined and is the cumulative proportion of the volume of rain water younger than age $\tau$ in outflow rivers. Moreover, the value of the event rain water threshold age also remains to be determined. Note that the event rain water threshold age is the intersection point of $\Omega_{Q1}$ and $\Omega_{Q2}$ (Figure 1),so we will derive $\Omega_{Q1}$ first. Then, the event rain water threshold age can be calculated.

**(1) Derivation of $\Omega_{Q1}$**

In the rainfall mixing model (line 282 and line 303), the volume of rain water aged $\tau$ in outflow rivers, i.e., $F_{out\_rain}(t,\tau)$ is tracked as:

$$F_{out\_rain}(t,\tau) = \overline{F_{out_T}}(t,\tau) \cdot f(t,\tau) = F_{out}(t)\,(1 - \Omega_Q) \cdot f(t,\tau) \tag{2}$$

where $\overline{F_{out_T}}(t,\tau)$ is the volume of water older than age $\tau$ in outflow rivers, $f(t,\tau)$ is the rainfall mixing factor. Let's divide $\Omega_{Q1}$ into two parts based on the event rain water threshold point: $\Omega_{Q1\_left}$ and $\Omega_{Q1\_right}$. For $\Omega_{Q1\_left}$, i.e., $\tau < \tau_e$, the water in outflow is all event rain water, so $\Omega_Q = \Omega_{Q1\_left}$. Substituting $\Omega_Q = \Omega_{Q1\_left}$ into equation (2), we have:

$$F_{out\_rain}(t,\tau) = F_{out}(t)\,(1 - \Omega_{Q1\_left}) \cdot f(t,\tau) \tag{3}$$

Then, $\Omega_{Q1\_left}$ can be calculated based on its definition and equation (3):

$$\Omega_{Q1\_left} = \int_0^\tau \frac{F_{out\_rain}(t,\tau)}{F_{out}(t)}\,d\tau = \int_0^\tau (1 - \Omega_{Q1\_left}) \cdot f(t,\tau)\,d\tau \tag{4}$$

By differentiating both sides of Equation (4) with respect to $\tau$, we obtain:

$$\frac{d\Omega_{Q1\_left}}{d\tau} = (1 - \Omega_{Q1\_left}) \cdot f(t,\tau) \tag{5}$$

Then, $\Omega_{Q1\_left}$ is solved from equation (5):

$$\Omega_{Q1\_left} = 1 - e^{-\int_0^\tau f(t,\tau)d\tau} \tag{6}$$

**(2) Determination of event rain water threshold age**

Since the event rain water threshold age $\tau_e$ is the intersection between $\Omega_{Q1\_left}$ and $\Omega_{Q2}$, this threshold age can be solved by setting $\Omega_{Q1\_left} = \Omega_{Q2}$, i.e.,

$$\Omega_{Q2} = 1 - e^{-\int_0^{\tau_e} f(t,\tau)d\tau} \tag{7}$$

**(3) Analytical form of the SAS function for outflow rivers:**

Finally, the analytical form of $\Omega_Q$ is:

$$\Omega_Q\big(t, S_T(t,\tau)\big) = \begin{cases} 1 - e^{-\int_0^\tau f(t,\tau)d\tau} & \text{if } \tau < \tau_e \\ \Omega_{Q2} & \text{if } \tau > \tau_e \end{cases} \tag{8}$$

where $\Omega_{Q2}$ is the cumulative beta distribution. ($\Omega_{Q1\_left}$ in equation (8) is actually the cumulative backward TTD for $\tau < \tau_e$. The transformation between the cumulative age-ranked SAS function and the cumulative backward TTD is ignored here, see Harman 2015, http://dx.doi.org/10.1002/2014WR015707.)

**(4) Compositions of water older than $\tau_e$**

For water older than $\tau_e$, $\Omega_{Q2}$, the beta distribution, is equal to the cumulative SAS function for all water (i.e., $\Omega_Q$). According to the rainfall mixing model, the cumulative proportion of rain water

$\Omega_{Q1\_right}$ is:

$$\Omega_{Q1\_right} = \int_{\tau_e}^{\tau}(1 - \Omega_{Q2}) \cdot f(t,\tau)\, d\tau + 1 - e^{-\int_0^{\tau_e} f(t,\tau)d\tau} \qquad (9)$$

and the cumulative proportion of river water in outflows is $\Omega_{Q2} - \Omega_{Q1\_right}$, which is shown in Figure 1.

**(5) Summary**

The reason we introduce a new SAS function is that the beta distribution may failed to quantify the proportion of event young rain water. In the system where the event young rain water exists, as shown in Figure 1, the proportion of event young rain water is always underestimated, if the SAS function for $\tau > \tau_e$ is used for the case of $\tau < \tau_e$. This new SAS function is in analytical form and will replace our previous numerical method for the SAS calculation. The modification of the SAS function will not change the results of TTD, RTD in Lake Taihu. The new model and its results, especially the value of event rain water threshold age, will be updated in our revised manuscript.

Below are our point-by-point replies:

This manuscript introduces a travel time model that deals with the problem of tracking water age in hydrologic systems characterised by multiple input fluxes. This is relevant for lake studies, which typically have 2 inputs (river inflow and rainfall). The authors apply the model to the case study of Lake Taihu, China, and calibrate the model parameters against 24 monthly values of mean isotope composition of the lake water. The authors use the calibrated model to discuss water age dynamics and rain/river water partitioning within the lake and the output fluxes.

I anticipate that, while I appreciate this work, I am afraid there are major technical issues in the solution proposed by the authors that prevents publication of the current version of the paper, but I think that similar results produced under alternative – and possibly simpler solutions may be worth of publication.

Reply: Huge thanks for your valuable suggestion, which will make our revised manuscript much better than the current version. After carefully consideration of your suggestion, we have put forward a simpler and new solution, which encapsulates the transport and mixing of rain water and river into a single SAS function. This new SAS function is introduced in our overall reply and will be added to our revised manuscript.

The authors touch on a very interesting and challenging problem that is multiple-input tracking in lumped models. Most of the water transit time literature (see the review work that myself and many colleagues have written in 2022, https://doi.org/10.1029/2022WR033096) dealt with systems with one single input, i.e. rainfall. Therefore, the effort put in place by the authors fills an important gap and is to be credited. The text, while needing some English proofread, is understandable and the approach undertaken by the authors appears rigorous and is described in full detail.

Reply: We appreciate your recognition of the contribution of our manuscript. Your acknowledgement has provided us with great encouragement to improve the TTD model in our

manuscript.

A key point of the paper is a new solution to the tracking of multiple sources within a lumped model. The authors call their solution "Rainfall mixing" or "rainfall tracking". I think this solution is problematic on some fronts:

1. The manuscript seems to often confound space with age. Lumped age models do not explicitly account for space. The effects of physical processes (like advection and dispersion) can be effectively reproduced by using a lagged transit time distribution or SAS function, but those processes are not directly modelled. Similarly, water age models do not account for any physical mixing within the storage and they only prescribe how the output fluxes remove waters of different ages from the storage (which is why the community tends to speak about "random sampling" rather than "well mixing"). The idea that rainfall water is well mixed with pre-existing lake water while river water is not is unsuitable to this lumped framework and should rather be translated into SAS language and equations. Translating the different transport processes of two very different inputs into a single SAS function may be challenging and highlights the limitations of lumped water age models.

Reply: We accept this comment. We will check through the manuscript to ensure the SAS language is consistently used when discussing our rainfall mixing model and the water age model. In the model development sections, where mixing in space is most often mentioned to discussing the rainfall mixing model, we will immediately translate these discussions into SAS language after these sentences. For example, in lines 239-240, after introducing the concept that rainwater is well mixed vertically with pre-existing lake water parcels, we will add the following sentence: "This indicates that the ratio between the volume of aged rainwater and the volume of lake water older than that rainwater remains fixed. Thus, in the outflow rivers, rainwater aged τ is randomly sampled from the mixed lake water older than age τ, and the random sampling probability or ratio in the outflow rivers is equal to the mixing ratio in the lake." In this way, the physical meaning of our model remains clear when we introduce our model in SAS language. This description approach is commonly used in SAS literature (Pangle et al. https://doi.org/10.1002/2016WR019901; Kim et al 2022, https://doi.org/10.1029/2020WR028959; Wilusz et al, https://doi.org/10.1029/2019WR025140; etc.).

Regarding the issue you mentioned in the last sentence, the new SAS function we proposed in our previous response has successfully integrated the transport of rain water and river water in the lake into a single SAS function. Therefore, this will not be an issue in our revised manuscript.

2. The rainfall mixing approach is difficult to understand and after reading it multiple times I still am not sure I could follow all the steps. My understanding is that ultimately the "candidate" beta-shaped SAS functions are checked at any time step and if some constrain is not respected, they are modified. The effect of this transformation is difficult to follow, but I think it generally increases the amount of young water released to the outflow, to effectively simulate the "preferential" release of rain water. The manuscript should clarify the effect of this modification more explicitly, for example by showing some examples of candidate and modified SAS

function or by showing a simulation with and without those constraints. I think it is also important that this more complex approach is justified in terms of model performance, i.e. that the model with the modification performs significantly better than a "traditional" model.

Reply: In our new SAS function, the steps for check the "candidate" beta-shaped SAS functions are canceled. The reason we need to check the beta-shaped SAS function is that it may fail to capture the proportion of event young rain water in outflow rivers, as shown in Figure 1. For detail, please see our overall reply at the beginning. In our new improved model, the SAS function is clearly defined in its analytical form, with no need to check the constrains. Therefore, this major issue will no longer be a problem in our revised manuscript.

3. Monthly time steps for the solution of the water age balance using the Euler Forward scheme is potentially coarse. I invite the authors to verify that the numerical accuracy is not compromised by the use of large time steps.

Reply: We accept this comment. A large Euler Forward scheme for partial differential equations may cause two problems: stability and accuracy. For the stability problem, this is not an issue, as our calculated TTD, RTD, and deuterium concentration do not blow up and remain within a reasonable range. However, we have not yet examined the accuracy issue as you suggested. We will add a discussion of model accuracy to the manuscript.

Below are the preliminary results of the verification of model accuracy as influenced by time step:

To verify the model accuracy, we interpolated our monthly lake volume data into datasets with several different time steps. For other input data, such as rainfall, ET, and river inflow and outflow rates, we continued using the monthly averaged data. We then reran our model with different time steps: dt=1, 0.5, 025, 0.125 months. The calculated deuterium concentrations are shown in Figure 2.

[Figure]

Figure 2. Influence of time steps on deuterium concentration calculation. dt=0.125 month is not displayed, as it is very close to the curve for dt=0.25.

In Figure 2, the deuterium concentration increases slightly with smaller time steps. Referring to Figure 13 in our manuscript, a larger time step may have the most significant influence on the estimation of evaporation fractionation factor for deuterium, but it has little effect on the parameters for the SAS function.

Therefore, we will revise our manuscript using the model with a smaller time step of dt=0.25 month

and add a paragraph to discuss the error caused by different time steps.

I believe these major issues need to be addressed before discussing additional minor comments. However, I reiterate that an improved or simplified multiple-input tracking system would likely make the paper worthy of publication.

Reply: With our newly developed SAS function, we believe the multi-input-tracking system will be much simpler and easier to understand and implement than our previous model. We also appreciate your comments on the time step, as they have significantly increased our model's accuracy.

Thank you again for your valuable comments; they will greatly improve the quality of our manuscript!